# FACTS: A Future-Aided Causal Teacher-Student Framework for Multimodal Time Series Forecasting

## Abstract

Traditional *unimodal* time series forecasting models often perform unreliably in real-world applications because they fail to capture the underlying causal drivers of temporal change. Fortunately, auxiliary modalities can unveil these drivers, *e.g.*, sky images capture the illumination conditions that govern solar power generation. However, the most informative *future* auxiliary signals directly tied to the target time series are unavailable at inference, while integrating such data is further hindered by cross-modal heterogeneity and structural mismatch. To address these challenges, we propose FACTS, a Future-Aided Causal Teacher-Student framework for *multimodal* time series forecasting. The teacher network, used only during training, leverages future auxiliary data to disentangle the causal responses underlying temporal dynamics, while the student network, trained solely on historical data, learns such causal knowledge via our proposed causal-perturbation contrastive distillation. To accommodate heterogeneous inputs, we design a bilinear orthogonal projector that efficiently converts high-dimensional auxiliary data into a compact series over time, allowing us to model both auxiliary data and time series via a unified bidirectional attention backbone. Furthermore, we devise a lag-aware fusion to align cross-modal signals within a tolerance window and apply random modality dropout to enhance the student's robustness to modality missingness. Extensive experiments on benchmark datasets demonstrate that FACTS significantly outperforms state-of-the-art methods, achieving average improvements of 32.98% in MSE and 22.25% in MAE. Code is available at https://anonymous.4open.science/r/FACTS-7F94.

## 1 Introduction

Time Series Forecasting (TSF) is a fundamental task in various real-world applications, including financial management (Elliott & Timmermann, 2013), energy consumption prediction (Trindade, 2015), and weather forecasting (Angryk et al., 2020). Recently, deep learning has driven rapid progress in TSF, existing methods mainly leverage Multi-Layer Perceptrons (MLPs) (Wang et al., 2024b), Recurrent Neural Networks (RNNs) (Lin et al., 2023), Convolutional Neural Networks (CNNs) (Wu et al., 2023), Transformers (Zhou et al., 2022), and Large Language Models (LLMs) (Jia et al., 2024) as backbones. By learning complex temporal patterns inherent in time series, these models often perform well on single-modality benchmark datasets.

Despite strong benchmark results, current *unimodal* TSF models frequently perform unreliably in practice because they fail to capture the underlying causal drivers of temporal dynamics (Melnychuk et al., 2022). For example, in solar power forecasting, when power output is rising during a sustained period of clear-sky conditions, these models tend to extrapolate continued growth across subsequent horizons. However, in real-world scenarios, sudden cloud cover may drastically reduce illumination conditions and cause a sharp drop in power generation. In such cases, models trained solely on time series are unable to understand how abrupt illumination changes affect the power generation, leading to unreliable predictions.

In many applications, rich data from other modalities (*i.e.*, *auxiliary modalities*) can reveal the causal drivers behind time series variation (Williams et al., 2024; Lee et al., 2025). For solar power fore-

casting, sky images provide real-time illumination cues that directly drive power output, whereas meteorological variables (*e.g.*, temperature and wind speed) potentially modulate irradiance and conversion efficiency. Effectively exploiting auxiliary data is promising to promote forecasting models to identify key drivers of change, thereby achieving more accurate and reliable predictions.

However, leveraging auxiliary data is nontrivial and poses three key challenges (Liu et al., 2025c; Ni et al., 2025a): (i) *Causal Unobservability*. While historical auxiliary signals can help models learn cross-modal dependencies and explain why the series changes, predictions remain vulnerable to abrupt environmental shifts. In contrast, future auxiliary signals can indicate how the future trajectory will evolve and thereby stabilize forecasts, but such information is unavailable at inference time. (ii) *Data Heterogeneity*. Time series data encodes continuous, evolving dynamics (*e.g.*, trends and periodicities), whereas auxiliary modalities such as images provide discrete scene snapshots (*e.g.*, sunny vs. cloudy). These semantic and temporal discrepancies hinder straightforward multimodal fusion. (iii) *Structural Mismatch*. Images are high-dimensional tensors, while meteorological measurements are low-dimensional vectors, which complicates multimodal architecture design. As a result, existing methods often struggle to effectively exploit auxiliary modalities to improve forecasting performance.

To address the above challenges, we propose FACTS, a Future-Aided Causal Teacher-Student framework for *multimodal* TSF. Our FACTS comprises a teacher network and a deployable student network that share a unified bidirectional-attention backbone. The teacher network ingests historical time series together with both historical and future auxiliary data to disentangle the causal drivers of unseen future time series. The student network is trained only on historical data and acquires meaningful causal knowledge from the teacher network via our Causal-Perturbation Contrastive Distillation (CPCD). To handle heterogeneous and structurally mismatched inputs, we introduce a Bilinear Orthogonal Projector (BOP) that maps auxiliary data to compact series over time. Accordingly, the teacher and student networks can employ the unified backbone to capture temporal dependencies from both auxiliary and temporal data and be trained end-to-end. Finally, we devise a lag-aware fusion mechanism to align temporal signals extracted from various modalities within a tolerance window to obtain the final forecasts. We also apply random modality dropout during student training to enhance its robustness to modality missingness caused by sensor outages or transmission interruptions. By effectively exploring multimodal causal drivers while distilling them into a purely historical student network, FACTS consistently achieves State-Of-The-Art (SOTA) performance across multiple datasets. The contributions of this paper are summarized as:

1. We propose FACTS, a novel multimodal TSF framework, which employs a future-aided teacher network to uncover causal drivers for the target series, while distilling them to promote the performance of a historical-only, deployable student network via CPCD.

2. We propose BOP that maps heterogeneous and shape-mismatched auxiliary data to compact serialized data, enabling a unified bidirectional-attention backbone across modalities.

3. We devise lag-aware multimodal fusion to align cross-modal signals within a tolerance window and introduce random modality dropout during student training to handle missing modalities, together improving the model's robustness in real-world scenarios.

4. Our FACTS consistently achieves SOTA performance across multiple real-world datasets, with average improvements of 32.98% in MSE and 22.25% in MAE.

## 2 RELATED WORK

### 2.1 UNIMODAL AND MULTIMODAL TIME SERIES FORECASTING

TSF has been extensively investigated in various domains like power systems (Trindade, 2015) and weather forecasting (Angryk et al., 2020). Traditional *unimodal* approaches rely solely on historical series and employ neural architectures such as MLPs (Wang et al., 2024b), RNNs (Lin et al., 2023), and CNNs (Wu et al., 2023) to capture temporal dependencies. Recently, Transformer-based methods (Zhou et al., 2021; Liu et al., 2024c) have achieved SOTA performance on public benchmarks. However, these models are trained on a single temporal modality and are blind to the causal drivers of time series, leaving them prone to sudden environmental changes in real-world scenarios.

To overcome the above limitations, *multimodal* approaches (Ni et al., 2025a; Skenderi et al., 2024; Shen et al., 2025) seek to enhance forecasting with auxiliary modalities. Prevailing methods mainly synthesize images or text from time series. Image-based methods (Liu et al., 2025b; Zhong et al., 2025) convert time series into line charts or spectrograms and then extract spatiotemporal features from them. Text-based methods (Jin et al., 2024; Cheng et al., 2024) map time series into textual data via tokenization, prompting, or reprogramming. Although these methods can strengthen the model's understanding of statistical regularities already present in time series, they do not introduce truly exogenous information that actually governs temporal dynamics. By contrast, recent studies (Jiang et al., 2025b; Ye et al., 2024) incorporate external signals. GPT4MTS (Jia et al., 2024) pairs time series with contemporaneous event descriptions via LLM-based summarization. VISUELLE (Skenderi et al., 2022) integrates product images with sales data to capture visual cues for demand forecasting. However, these approaches associate an entire time series with a single static description or image, which fails to capture evolving factors that influence temporal dynamics, and still achieve limited performance.

In this paper, we consider time-varying auxiliary data accompanying each timestamp. To the best of our knowledge, we are the first to formulate and study this per-timestep multimodal TSF setting. By effectively processing and aligning these auxiliary signals with time series, our method captures evolving causal drivers behind temporal dynamics, and thus enabling reliable forecasting.

## 2.2 AUXILIARY–TEMPORAL MODALITY ALIGNMENT

Auxiliary modalities differ fundamentally from time series. To bridge this gap, existing works (Xue & Salim, 2023; Liu et al., 2024b) align multimodal data at the input or representation levels. Prompt-Cast (Xue & Salim, 2023) encodes time series and generated text into a single prompt as model input. TimeLLM (Jin et al., 2024) embeds dataset descriptions as semantic prototypes and concatenates them with temporal representations. CALF (Liu et al., 2024b) enforces cross-branch consistency between temporal and textual pathways at both intermediate and output layers. TS-TCD (Wang et al., 2024a) uses self-attention to align time series with word embeddings learned by the LLM. These methods often construct auxiliary data from the time series itself and without introducing any external information, so no cross-modal temporal misalignment arises. In practice, however, multimodal signals exhibit inherent temporal lags (see App. B.2), which are often overlooked by existing methods. Therefore, we propose a lag-aware fusion mechanism that computes cross-modal similarities within a tolerance window, thereby improving predictive reliability.

## 2.3 CAUSAL LEARNING AND KNOWLEDGE DISTILLATION

Causal learning (Melnychuk et al., 2022; Gopnik et al., 2004) estimates causal effects among variables to promote reliable inference. In general, existing methods model causal relations via causal graph construction (Wei et al., 2022), invariant representation learning (Deng & Zhang, 2021), or counterfactual reasoning (Melnychuk et al., 2022). Among them, counterfactual-based methods are rather simple and effective, which alter specific variables and assess their impact on outcomes. For example, Causal Transformer (Melnychuk et al., 2022) generates counterfactual time series and employs a three-branch attention architecture to jointly model treatments, confounders, and outcomes. DAG-Aware Transformer (Liu et al., 2024a) separately computes observations and counterfactual outcomes for intervention and non-intervention groups, and estimates their differences.

Knowledge Distillation (KD) (Gou et al., 2021; Cho & Hariharan, 2019) transfers knowledge from a pretrained teacher network to improve the performance of a student network. TimeDistill (Ni et al., 2025b) extracts multiscale temporal patterns from a complex Transformer to guide a lightweight MLP. TimeKD (Liu et al., 2025a) leverages a teacher network with access to single-modality future time series to generate high-quality features and transfer them to the student network via feature alignment. In this work, we integrate causal learning with knowledge distillation and propose CPCD. The teacher network ingests real and perturbed future multimodal auxiliary data, which is trained to capture multimodal causal dependencies. The historical-only student network is promoted to learn faithful causal knowledge from the teacher network while apart from perturbed causal representations through a contrastive objective, thereby improving forecasting performance.

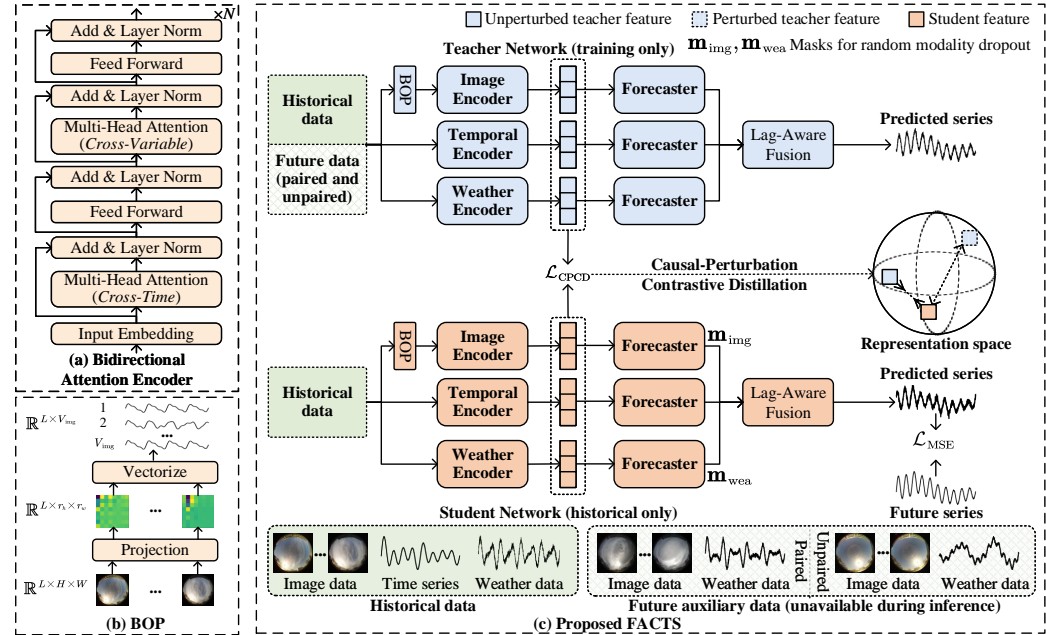

Figure 1: Overview of FACTS. (a) Bidirectional attention encoder for modality-specific branches. (b) Bilinear Orthogonal Projector (BOP) that maps images to a compact multivariate series. (c) *Teacher network* (blue) utilizes historical data and paired/unpaired future auxiliary inputs to produce *unperturbed* and *perturbed* features that contain faithful and spurious causal drivers, respectively. *Student network* (orange) encodes historical multimodal inputs to forecast the future series. Meanwhile, causal-perturbation contrastive distillation pulls the student feature toward the teacher's unperturbed feature and away from perturbed ones, thereby learning meaningful causal knowledge. *Note:* Only the student network and historical data are used for inference.

## 3 APPROACH

Traditional *unimodal* TSF predicts future series $\mathbf{y} \in \mathbb{R}^{T \times V_{\text{time}}}$ over a horizon $T$ from a historical series $\mathbf{x}_{\text{time}} \in \mathbb{R}^{L \times V_{\text{time}}}$ with $L$ steps, where $V_{\text{time}}$ is the number of temporal variables. In this paper, our FACTS focuses on practical *multimodal* scenarios and incorporates auxiliary modalities to capture causal factors of temporal dynamics, thereby achieving accurate and reliable forecasts. For example, in solar power forecasting, each time series $\mathbf{x}_{\text{time}}$ is accompanied by historical images $\mathbf{x}_{\text{img}} \in \mathbb{R}^{L \times H \times W}$ (channel dimension omitted) and weather data $\mathbf{x}_{\text{wea}} \in \mathbb{R}^{L \times V_{\text{wea}}}$, as well as future images $\mathbf{y}_{\text{img}} \in \mathbb{R}^{T \times H \times W}$ and weather data $\mathbf{y}_{\text{wea}} \in \mathbb{R}^{T \times V_{\text{wea}}}$.

As illustrated in Fig. 1, our approach comprises three components. First, we introduce BOP to unify heterogeneous multimodal data (Sec. 3.1), which converts high-dimensional auxiliary data into a serialized format. This enables our FACTS to utilize a unified backbone with bidirectional attention to capture both temporal and spatial dependencies across all modalities (Sec. 3.1). Second, we propose lag-aware fusion and random modality dropout, which explicitly address potential cross-modal temporal misalignment and modality missingness in practice, thereby effectively combining predictions from modality-specific branches (Sec. 3.2). Third, we employ a teacher network to grasp causal drivers from future auxiliary data and transfer them to improve the performance of the student network via our proposed CPCD, as detailed in Sec. 3.3.

### 3.1 UNIFIED MULTIMODAL PROCESSING VIA DATA SERIALIZATION

In real-world scenarios, rich auxiliary data is available to strengthen TSF. Existing methods (Jin et al., 2024; Zhong et al., 2025) usually encode auxiliary data with models pretrained on image or textual data. However, time series and auxiliary data are heterogeneous and structurally mismatched (Nie et al., 2023b). These pre-trained models struggle to effectively capture meaningful

temporal dynamics from auxiliary data. Meanwhile, cross-modal heterogeneity demands tailored branches for each modality, which complicates system design. Moreover, auxiliary data (*e.g.*, images) are far higher-dimensional than time series, inflating compute costs. To address these issues, we introduce a Bilinear Orthogonal Projector (BOP) that directly maps high-dimensional auxiliary data to low-dimensional, ready-to-use serialized data. As a result, our FACTS can employ a unified backbone to effectively and efficiently process various modalities.

**Bilinear Orthogonal Projector.** Given a high-dimensional image sequence $\mathbf{x}_{\text{img}} \in \mathbb{R}^{L \times H \times W}$, we perform frame-wise dimensionality reduction with a *learnable* bilinear projector. Let $\mathbf{U} \in \mathbb{R}^{H \times r_h}$ and $\mathbf{V} \in \mathbb{R}^{W \times r_w}$ ($r_h \ll H$, $r_w \ll W$) be the row and column projection matrices, respectively, we process each frame $\mathbf{x}_{\text{img},l} \in \mathbb{R}^{H \times W}$ as:

$$\mathbf{x}_{\text{img},l}^{(\text{BOP})} = \mathbf{U}^\top \mathbf{x}_{\text{img},l} \mathbf{V} \in \mathbb{R}^{r_h \times r_w}, \qquad l = 1, \ldots, L. \tag{1}$$

During training, $\mathbf{U}$ and $\mathbf{V}$ are regularized toward orthogonality via $\|\mathbf{U}^\top \mathbf{U} - \mathbf{I}\|_F^2 + \|\mathbf{V}^\top \mathbf{V} - \mathbf{I}\|_F^2$, where $\mathbf{I}$ is the identity matrix. Then, we vectorize each projected frame as:

$$\mathbf{x}_{\text{img},l}^{(\text{vec})} = \text{vec}\big(\mathbf{x}_{\text{img},l}^{(\text{BOP})}\big) \in \mathbb{R}^{V_{\text{img}}}, \quad V_{\text{img}} := r_h r_w, \quad l = 1, \ldots, L. \tag{2}$$

Finally, we stack these vectors over time to form a multivariate series with $V_{\text{img}}$ variables, as follows:

$$\mathbf{x}_{\text{img}}^{(\text{vec})} = \begin{bmatrix} \mathbf{x}_{\text{img},1}^{(\text{vec})} & \cdots & \mathbf{x}_{\text{img},L}^{(\text{vec})} \end{bmatrix}^\top \in \mathbb{R}^{L \times V_{\text{img}}}. \tag{3}$$

In $\mathbf{x}_{\text{img}}^{(\text{vec})} \in \mathbb{R}^{L \times V_{\text{img}}}$, the $v$-th column $\big(\mathbf{x}_{\text{img}}^{(\text{vec})}\big)_{:,\,v} \in \mathbb{R}^L$ forms a univariate series describing the dynamics at spatial location $v$, the $l$-th row $\big(\mathbf{x}_{\text{img}}^{(\text{vec})}\big)_{l,\,:} \in \mathbb{R}^{V_{\text{img}}}$ summarizes the spatial state at time $l$. Inspired by (Zhang & Yan, 2023), we employ a bidirectional attention backbone to capture temporal and spatial dependencies from both temporal and auxiliary series. We adopt BOP rather than classical downsampling methods (Abdi & Williams, 2010; Stewart, 1993) as it preserves 2D spatial structure while only incurring negligible overhead, analyzed in App. A.1. Last but not least, BOP requires no offline precomputation and is trained *end-to-end* jointly with the forecasting backbone.

**Backbone with Bidirectional Attention.** We employ a backbone consisting of an encoder $f_\theta$ with $N$ bidirectional-attention blocks and an MLP forecaster $g_\phi$. The encoder extracts temporal and spatial dependencies from both temporal and auxiliary inputs, and the forecaster maps the encoder's hidden embedding $\mathbf{h}_{\text{m}} \in \mathbb{R}^{P \times V_{\text{m}} \times D}$ to the future series $\hat{\mathbf{y}}_{\text{m}} \in \mathbb{R}^{T \times V_{\text{time}}}$:

$$\mathbf{h}_{\text{m}} = f_{\theta_{\text{m}}}(\mathbf{x}_{\text{m}}), \quad \hat{\mathbf{y}}_{\text{m}} = g_{\phi_{\text{m}}}(\mathbf{h}_{\text{m}}), \tag{4}$$

where $\text{m} \in \{\text{time}, \text{img}, \text{wea}\}$, for simplicity, we omit modality-specific superscript/subscript below.

Specifically, we first patch the input series $\mathbf{x} \in \mathbb{R}^{L \times V}$ into a temporal embedding $\mathbf{z} \in \mathbb{R}^{P \times V \times D}$, where $P$ is the number of patches and $D$ is the per-patch embedding dimension (detailed in App. A.2). In each bidirectional-attention block, we process its embedding sequentially with cross-time attention and cross-variable attention.

*Cross-Time Attention.* For each variable $v \in \{1, \ldots, V\}$, we apply Multi-Head Self-Attention (MHSA) over the temporal dimension (length $P$) to capture temporal dependencies:

$$\mathbf{z}_{:,v,:}^{(n,1)} = \text{LayerNorm}(\mathbf{z}_{:,v,:}^{(n,0)} + \text{MHSA}_{\text{time}}(\mathbf{z}_{:,v,:}^{(n,0)})), \tag{5}$$

$$\mathbf{z}_{:,v,:}^{(n,2)} = \text{LayerNorm}(\mathbf{z}_{:,v,:}^{(n,1)} + \text{MLP}_{\text{time}}(\mathbf{z}_{:,v,:}^{(n,1)})). \tag{6}$$

Here, $n \in \{1, \ldots, N\}$ indexes blocks, we set $\mathbf{z}^{(1,0)} = \mathbf{z}$ and define $\mathbf{z}^{(n+1,0)} = \mathbf{z}^{(n,4)}$ (for $n < N$).

*Cross-Variable Attention.* For each patch with index $p \in \{1, \ldots, P\}$, we also apply MHSA over the variable dimension (length $V$) to model inter-variable dependencies:

$$\mathbf{z}_{p,:,:}^{(n,3)} = \text{LayerNorm}(\mathbf{z}_{p,:,:}^{(n,2)} + \text{MHSA}_{\text{var}}(\mathbf{z}_{p,:,:}^{(n,2)})) \tag{7}$$

$$\mathbf{z}_{p,:,:}^{(n,4)} = \text{LayerNorm}(\mathbf{z}_{p,:,:}^{(n,3)} + \text{MLP}_{\text{var}}(\mathbf{z}_{p,:,:}^{(n,3)})). \tag{8}$$

The output of the $N$-th block is the final hidden embedding $\mathbf{h} = \mathbf{z}^{(N,4)}$ for the encoder. Then, the forecaster maps the hidden embedding as the predicted future series, see App. A.2.

## 3.2 ROBUST AND LAG-AWARE MULTIMODAL FUSION

In this paper, we focus on practical multimodal scenarios, where auxiliary modalities are available and can be utilized to improve TSF. As shown in Fig. 1, the multimodal teacher/student network comprises multiple branches (temporal, image, and weather), where each branch is instantiated with the unified bidirectional-attention backbone and modality-specific configurations (provided in App. A.3). Here, we illustrate the multimodal fusion using the student network, the teacher network that accesses future auxiliary data follows the same fusion steps. Specifically, time series $\mathbf{x}_{\text{time}} \in \mathbb{R}^{L \times V_{\text{time}}}$, images $\mathbf{x}_{\text{img}} \in \mathbb{R}^{L \times V_{\text{img}}}$, and weather records $\mathbf{x}_{\text{wea}} \in \mathbb{R}^{L \times V_{\text{wea}}}$ are processed by their respective branches to predict the future time series as follows:

$$\hat{\mathbf{y}}_{\text{time}} = g_{\phi_{\text{time}}}(f_{\theta_{\text{time}}}(\mathbf{x}_{\text{time}})), \quad \hat{\mathbf{y}}_{\text{img}} = g_{\phi_{\text{img}}}(f_{\theta_{\text{img}}}(\mathcal{B}(\mathbf{x}_{\text{img}}))), \quad \hat{\mathbf{y}}_{\text{wea}} = g_{\phi_{\text{wea}}}(f_{\theta_{\text{wea}}}(\mathbf{x}_{\text{wea}})), \quad (9)$$

where $\mathcal{B}$ represents the BOP, $\hat{\mathbf{y}}_{\text{time}}, \hat{\mathbf{y}}_{\text{img}}, \hat{\mathbf{y}}_{\text{wea}} \in \mathbb{R}^{T \times V_{\text{time}}}$.

**Lag-Aware Multimodal Fusion.** In practice, temporal misalignment often occurs across modalities (Nie et al., 2023b; 2024), as visualized in Fig. A-1. For example, when a cloud nears a solar power station, the sky camera can detect it immediately, but the power series drops only after it actually shades the panels. To handle such offsets, we combine modality-specific predictions using similarities computed within a lag window. Given the nonnegative lag set $\{0, 1, \ldots, \delta_{\max}\}$, the variable-wise similarities between temporal and image modalities are calculated as:

$$s_{\text{img},\delta}^{(v)} = \sum_{t=1}^{T-\delta} \hat{\mathbf{y}}_{\text{time}, t+\delta, v} \, \hat{\mathbf{y}}_{\text{img}, t, v}, \quad s_{\text{img}}^{(v)} = \max(s_{\text{img},\delta}^{(v)}), \quad \delta \in \{0, 1, \ldots, \delta_{\max}\}. \quad (10)$$

By repeating the similarity computation across $V_{\text{time}}$ predicted variables, we obtain the similarities between the temporal and image data as $\mathbf{s}_{\text{img}} = \left[ s_{\text{img}}^{(1)}, \ldots, s_{\text{img}}^{(V_{\text{time}})} \right]^{\top} \in \mathbb{R}^{V_{\text{time}}}$. Meanwhile, the similarities $\mathbf{s}_{\text{wea}} \in \mathbb{R}^{V_{\text{time}}}$ between temporal and weather data are calculated in the same way as $\mathbf{s}_{\text{img}}$.

**Random Modality Dropout.** Auxiliary data may be unavailable in practice due to sensor failures or transmission interruptions (Wu et al., 2024; Jiang et al., 2025a), which renders the corresponding auxiliary branches inoperative and degrades the forecast reliability of the student network. To mitigate such modality missingness, during student training, we apply stochastic masks on modality-specific predicted series. Accordingly, we fuse these modality-specific time series as follows:

$$\hat{\mathbf{y}} = \hat{\mathbf{y}}_{\text{time}} + (\mathbf{m}_{\text{img}} \odot \hat{\mathbf{y}}_{\text{img}}) \odot (\mathbf{1}\mathbf{s}_{\text{img}}^{\top}) + (\mathbf{m}_{\text{wea}} \odot \hat{\mathbf{y}}_{\text{wea}}) \odot (\mathbf{1}\mathbf{s}_{\text{wea}}^{\top}). \quad (11)$$

Here, $\mathbf{1} \in \mathbb{R}^{T \times 1}$ denotes the all-ones column vector, which broadcasts $\mathbf{s}_{\text{img}}$ (resp. $\mathbf{s}_{\text{wea}}$) to $\mathbb{R}^{T \times V_{\text{time}}}$, and masks $\mathbf{m}_{\text{img}}, \mathbf{m}_{\text{wea}} \in \{0, 1\}^{T \times V_{\text{time}}}$ are generated i.i.d. from $\text{Bernoulli}(\alpha)$. If a modality is missing at inference, its mask is set to $0$ (and to $1$ otherwise). The effectiveness of random modality dropout is analyzed in App. C.3. Finally, the student network is promoted to predict as accurately as the real future series $\mathbf{y}$ by minimizing the Mean Squared Error (MSE) loss:

$$\mathcal{L}_{\text{MSE}} = \frac{1}{BTV_{\text{time}}} \sum_{i=1}^{B} \sum_{t=1}^{T} \sum_{v=1}^{V_{\text{time}}} \left( \mathbf{y}_{t,v}^{i} - \hat{\mathbf{y}}_{t,v}^{i} \right)^2, \quad (12)$$

where $B$ denotes the batch size.

## 3.3 CAUSAL-PERTURBATION CONTRASTIVE DISTILLATION

Given historical time series and auxiliary data, the multimodal model can learn cross-modal dependencies and understand why the series changes, thereby improving performance. In practice, however, abrupt events (*e.g.*, rapid cloud occlusion) often arise without clear precursors (Toller et al., 2025; Zheng & Hu, 2022), and models trained solely on historical observations struggle to produce reliable forecasts. While future auxiliary signals can reveal imminent causal drivers of the target future series, they are unavailable at deployment. Therefore, we exploit future auxiliary signals during training without sacrificing deployability via a teacher-student framework. The teacher network ingests historical time series with historical and future auxiliary data to explore causal responses for temporal dynamics, while the historical-only student network learns the teacher's causal knowledge via the proposed CPCD. Notably, only the student network is retained for inference.

Table 1: Results of time series forecasting. FACTS achieves an average improvement of **32.98%** in MSE and **22.25%** in MAE. The best results are in **bold** while the second best are underlined. *Note:* all Standard Deviation (STD) values in the table are scaled by $\times 10^{-2}$, and the teacher network used only for distillation during training is excluded from method ranking.

| Modality | Model | Folsom | | | | SKIPP'D | | | | CCG | | | | CRNN | | | |
|---|---|---|---|---|---|---|---|---|---|---|---|---|---|---|---|---|---|
| | | MSE | STD | MAE | STD | MSE | STD | MAE | STD | MSE | STD | MAE | STD | MSE | STD | MAE | STD |
| Unimodal (Traditional) | Autoformer | 0.1655 | 1.43 | 0.2030 | 0.91 | 0.4012 | 1.41 | 0.4798 | 1.23 | 0.0051 | 0.02 | 0.0488 | 0.10 | 0.3677 | 17.68 | 0.538 | 13.92 |
| | Crossformer | 0.1386 | 0.26 | 0.2556 | 0.60 | 0.3322 | 0.43 | 0.4214 | 1.15 | 0.0060 | 0.07 | 0.0524 | 0.41 | 0.3114 | 14.94 | 0.4524 | 12.87 |
| | DLinear | 0.1886 | 0.24 | 0.2792 | 0.45 | 0.3410 | 0.82 | 0.4286 | 1.18 | 0.0034 | 0.01 | 0.0325 | 0.02 | 0.1667 | 0.12 | 0.3101 | 0.16 |
| | FEDformer | 0.0794 | 0.12 | 0.1172 | 0.36 | 0.3449 | 0.72 | 0.4376 | 0.74 | 0.0036 | 0.01 | 0.0343 | 0.03 | 0.3278 | 18.82 | 0.4709 | 13.06 |
| | Informer | 0.1218 | 0.04 | 0.1253 | 0.13 | 0.3305 | 0.48 | 0.4306 | 0.34 | 0.0044 | 0.03 | 0.0422 | 0.06 | 0.2044 | 1.35 | 0.3470 | 1.55 |
| | iTransformer | 0.1217 | 0.07 | 0.1115 | 0.07 | 0.3057 | 0.19 | 0.4219 | 0.13 | 0.0045 | 0.08 | 0.0434 | 0.64 | 0.1584 | 0.17 | 0.3105 | 0.13 |
| | MICN | 0.1451 | 0.08 | 0.1199 | 0.21 | 0.3445 | 0.30 | 0.4318 | 0.16 | 0.0033 | 0.01 | 0.0322 | 0.01 | 0.1998 | 1.40 | 0.3379 | 1.27 |
| | SegRNN | 0.0828 | 0.23 | 0.1199 | 1.27 | 0.3368 | 0.24 | 0.4244 | 0.21 | 0.0033 | 0.01 | 0.0319 | 0.02 | 0.1726 | 0.58 | 0.3087 | 0.71 |
| | TiDE | 0.2242 | 0.59 | 0.2695 | 0.65 | 0.3214 | 0.16 | 0.4361 | 0.03 | 0.0040 | 0.01 | 0.0362 | 0.02 | 0.1825 | 0.32 | 0.3221 | 0.68 |
| | TimesNet | 0.0937 | 0.31 | 0.1435 | 0.44 | 0.3208 | 0.49 | 0.4106 | 0.42 | 0.0036 | 0.01 | 0.0350 | 0.11 | 0.2010 | 1.76 | 0.3451 | 1.74 |
| | TimeXer | 0.0825 | 0.06 | 0.1151 | 0.24 | 0.3151 | 0.23 | 0.4076 | 0.24 | 0.0038 | 0.01 | 0.0391 | 0.10 | 0.1907 | 0.41 | 0.3327 | 0.54 |
| | TimeMixer | 0.0896 | 0.13 | 0.1282 | 0.35 | 0.3236 | 0.47 | 0.4188 | 0.53 | 0.0078 | 0.09 | 0.0685 | 0.14 | 0.1837 | 0.69 | 0.3270 | 0.65 |
| Unimodal (LLM-based) | CALF | 0.0853 | 0.01 | 0.1228 | 0.18 | 0.3131 | 0.43 | 0.4322 | 0.06 | 0.0036 | 0.01 | 0.0325 | 0.02 | 0.1710 | 0.05 | 0.3113 | 0.54 |
| | OFA | 0.2026 | 0.37 | 0.2101 | 0.22 | 0.3322 | 0.25 | 0.4202 | 0.13 | 0.0059 | 0.05 | 0.0539 | 0.23 | 0.2051 | 0.66 | 0.3517 | 0.56 |
| | LLMMixer | 0.1014 | 0.13 | 0.1456 | 0.37 | 0.3196 | 1.36 | 0.4337 | 0.82 | 0.0046 | 0.02 | 0.0356 | 0.17 | 0.2069 | 1.34 | 0.3512 | 1.04 |
| MultiModal | AimTS | 0.1366 | 1.26 | 0.1843 | 1.62 | 0.3025 | 0.21 | 0.4154 | 0.19 | 0.0042 | 0.01 | 0.0328 | 0.03 | 0.1774 | 0.14 | 0.3193 | 0.11 |
| | GPT4MTS | 0.1024 | 0.40 | 0.1440 | 0.53 | 0.3230 | 0.26 | 0.4039 | 0.27 | 0.0057 | 0.05 | 0.0326 | 0.08 | 0.1643 | 0.17 | 0.3064 | 0.10 |
| | TimeVLM | 0.1210 | 0.15 | 0.1599 | 0.24 | 0.3169 | 0.24 | 0.3966 | 0.19 | 0.0043 | 0.01 | 0.0317 | 0.01 | 0.1662 | 0.49 | 0.3026 | 0.51 |
| | FACTS (Student) | 0.0716 | 0.03 | 0.0968 | 0.05 | 0.2876 | 0.19 | 0.3843 | 0.23 | 0.0028 | 0.01 | 0.0315 | 0.01 | 0.1121 | 0.16 | 0.2497 | 0.06 |
| | FACTS (Teacher) | 0.0565 | 0.02 | 0.0923 | 0.07 | 0.2812 | 0.17 | 0.3701 | 0.15 | 0.0024 | 0.01 | 0.0303 | 0.01 | 0.1107 | 0.16 | 0.2466 | 0.04 |

For each time series $\mathbf{x}^i_{\text{time}}$, we form an *unperturbed* pair $(\mathbf{x}^i_{\text{time}}, \mathbf{x}^i_{\text{img}}, \mathbf{x}^i_{\text{wea}}, \mathbf{y}^i_{\text{img}}, \mathbf{y}^i_{\text{wea}})$ with matched historical and future auxiliary data, and a *perturbed* pair $(\mathbf{x}^i_{\text{time}}, \mathbf{x}^i_{\text{img}}, \mathbf{x}^i_{\text{wea}}, \mathbf{y}^j_{\text{img}}, \mathbf{y}^j_{\text{wea}})$ where future auxiliary data is replaced by random examples in the same batch ($i \neq j$). Both pairs are passed through the teacher network to produce unperturbed feature $\mathbf{h}^i_T$ and perturbed feature $\tilde{\mathbf{h}}^i_T$ as:

$$\mathbf{h}^i_T = \text{MLP}^T(\text{concat}(f^T_{\theta_{\text{time}}}(\mathbf{x}^i_{\text{time}}), f^T_{\theta_{\text{img}}}(\mathcal{B}^T(\text{concat}(\mathbf{x}^i_{\text{img}}, \mathbf{y}^i_{\text{img}}))), f^T_{\theta_{\text{wea}}}(\text{concat}(\mathbf{x}^i_{\text{wea}}, \mathbf{y}^i_{\text{wea}})))), \quad (13)$$

$$\tilde{\mathbf{h}}^i_T = \text{MLP}^T(\text{concat}(f^T_{\theta_{\text{time}}}(\mathbf{x}^i_{\text{time}}), f^T_{\theta_{\text{img}}}(\mathcal{B}^T(\text{concat}(\mathbf{x}^i_{\text{img}}, \mathbf{y}^j_{\text{img}}))), f^T_{\theta_{\text{wea}}}(\text{concat}(\mathbf{x}^i_{\text{wea}}, \mathbf{y}^j_{\text{wea}})))). \quad (14)$$

In parallel, the student network only ingests the historical inputs and its feature $\mathbf{h}^i_S$ is obtained as:

$$\mathbf{h}^i_S = \text{MLP}^S(\text{concat}(f^S_{\theta_{\text{time}}}(\mathbf{x}^i_{\text{time}}), f^S_{\theta_{\text{img}}}(\mathcal{B}^S(\mathbf{x}^i_{\text{img}})), f^S_{\theta_{\text{wea}}}(\mathbf{x}^i_{\text{wea}}))). \quad (15)$$

Here, superscripts (subscripts) '$T$' and '$S$' denote the teacher and student networks, respectively. Then, the CPCD loss is computed as follows:

$$\mathcal{L}_{\text{CPCD}} = -\frac{1}{B} \sum_{i=1}^{B} \log \frac{\exp\left(\left(\mathbf{h}^i_S\right)^\top \mathbf{h}^i_T / \tau\right)}{\sum_{j=1, j\neq i}^{B} \exp\left(\left(\mathbf{h}^i_S\right)^\top \mathbf{h}^j_T / \tau\right) + \sum_{k=1}^{B} \exp\left(\left(\mathbf{h}^i_S\right)^\top \tilde{\mathbf{h}}^k_T / \tau\right)}. \quad (16)$$

Here, $\tau$ is the temperature, for clarity, we omit feature normalization in the notation. By minimizing $\mathcal{L}_{\text{CPCD}}$, the student's features are pulled toward the teacher's unperturbed features and away from the teacher's perturbed features (visualized in App. C.6), thereby learning the causal drivers from the teacher network. Consequently, even without access to future auxiliary data, the student network can still yield reliable predictive performance. The total objective function of the student network is defined as:

$$\mathcal{L}_{\text{Total}} = \mathcal{L}_{\text{MSE}} + \lambda \mathcal{L}_{\text{CPCD}}, \quad (17)$$

where $\lambda > 0$ is a trade-off parameter to balance the contribution of $\mathcal{L}_{\text{MSE}}$ and $\mathcal{L}_{\text{CPCD}}$. Teacher network (with access to future auxiliary data) is trained only using MSE loss (detailed in App. A.4). To ensure teacher network acquires sufficient representational capacity to extract reliable causal signals for student network, we train it to convergence before student network distillation.

## 4 EXPERIMENTS

### 4.1 EXPERIMENTAL SETUP

**Datasets.** We study multimodal TSF, where each input pair consists of a target time series and auxiliary modalities (*e.g.*, images, weather records). We evaluate the proposed FACTS on two public solar power generation datasets, *i.e.*, Folsom (Pedro et al., 2019) and SKIPP'D (Nie et al., 2023a), and two water-level monitoring datasets, *i.e.*, CCG and CRNN.

Table 2: Results of component-wise model analysis, each row group replaces the indicated component of FACTS with alternatives while keeping all other parts unchanged. Error increases in MSE/MAE (lower is better) attributed to applied alternatives are denoted in red font in brackets. *Note:* all STD values in the table are scaled by $\times 10^{-2}$.

| Analysis Components | Algorithm | Folsom | | | | SKIPP'D | | | |
|---|---|---|---|---|---|---|---|---|---|
| | | MSE | STD | MAE | STD | MSE | STD | MAE | STD |
| CPCD | FKD | 0.0868 (↑ 21.22%) | 0.22 | 0.1338 (↑ 38.22%) | 0.71 | 0.3334 (↑ 18.56%) | 0.43 | 0.4168 (↑ 8.45%) | 0.68 |
| | CRD | 0.0818 (↑ 14.24%) | 0.16 | 0.1237 (↑ 27.78%) | 0.64 | 0.3250 (↑ 15.57%) | 0.68 | 0.4180 (↑ 8.76%) | 0.83 |
| | TimeKD | 0.0833 (↑ 16.34%) | 0.25 | 0.1195 (↑ 23.45%) | 0.46 | 0.3117 (↑ 10.84%) | 0.23 | 0.4163 (↑ 8.32%) | 0.70 |
| | TimeDistill | 0.0779 (↑ 8.79%) | 0.08 | 0.1121 (↑ 15.80%) | 0.53 | 0.3125 (↑ 11.13%) | 0.38 | 0.4051 (↑ 5.41%) | 0.55 |
| Multimodal Data Fusion | Gating | 0.0860 (↑ 20.11%) | 0.26 | 0.1331 (↑ 37.50%) | 1.37 | 0.3031 (↑ 7.78%) | 0.35 | 0.4035 (↑ 4.99%) | 0.48 |
| | Self-Attention | 0.0823 (↑ 14.94%) | 0.18 | 0.1262 (↑ 30.37%) | 1.44 | 0.2975 (↑ 5.79%) | 0.15 | 0.3992 (↑ 3.87%) | 0.27 |
| | Channel-Similarity | 0.0811 (↑ 13.26%) | 0.15 | 0.1199 (↑ 23.86%) | 1.16 | 0.2928 (↑ 4.13%) | 0.34 | 0.3944 (↑ 2.62%) | 0.40 |
| Backbone | iTransformer | 0.0867 (↑ 21.08%) | 0.32 | 0.1071 (↑ 10.64%) | 0.29 | 0.3149 (↑ 11.98%) | 0.26 | 0.4162 (↑ 8.30%) | 0.15 |
| | TimeMixer | 0.1142 (↑ 59.49%) | 0.65 | 0.1393 (↑ 43.90%) | 0.95 | 0.3012 (↑ 7.11%) | 0.29 | 0.3970 (↑ 3.30%) | 0.47 |
| | TimesNet | 0.0928 (↑ 29.61%) | 0.24 | 0.1149 (↑ 18.69%) | 0.52 | 0.3138 (↑ 11.59%) | 0.24 | 0.4059 (↑ 5.62%) | 0.35 |
| | GPT2 | 0.0968 (↑ 35.19%) | 0.30 | 0.1283 (↑ 22.21%) | 0.46 | 0.3175 (↑ 12.91%) | 0.49 | 0.4148 (↑ 7.94%) | 0.51 |
| | Llama-7B | 0.0867 (↑ 21.08%) | 0.26 | 0.1109 (↑ 14.56%) | 0.43 | 0.3064 (↑ 8.96%) | 0.44 | 0.4052 (↑ 5.43%) | 0.42 |
| **Ours** | **FACTS** | **0.0716** | 0.03 | **0.0968** | 0.05 | **0.2812** | 0.17 | **0.3843** | 0.23 |

**Baselines.** To assess the performance of FACTS, we compare it against representative methods, categorized as follows: (i) traditional unimodal methods, including MLP-based (Zeng et al., 2023; Das et al., 2023; Wang et al., 2024b), RNN-based (Lin et al., 2023), CNN-based (Wang et al., 2022; Wu et al., 2023), and Transformer-based (Wang et al., 2024c; Wu et al., 2021; Zhang & Yan, 2023; Zhou et al., 2022; Liu et al., 2022; Zhou et al., 2021; Liu et al., 2024c; Wang et al., 2024c) methods; (ii) LLM-based unimodal methods (Liu et al., 2024b; Zhou et al., 2023; Kowsher et al., 2024); and (iii) multimodal methods (Chen et al., 2025; Jia et al., 2024; Jin et al., 2024; Zhong et al., 2025). Further details regarding the datasets and baselines are provided in App. B.

**Implementation Details.** We use the Adam optimizer for both the teacher and student networks, with a learning rate of 0.001 and weight decay of 0.05. The modality-dropout probability is 0.4, the trade-off parameter $\lambda$ in Eq. (17) is 0.1, $\delta_{\max}$ in lag-aware multimodal fusion is 5, $r_h$ and $r_w$ in BOP are set to 8, these hyperparameters are analyzed in Sec. 4.4. To ensure statistical reliability, we repeat each experiment three times and report the mean and standard deviation.

### 4.2 EXPERIMENTAL RESULTS

**Settings.** We conducted extensive experiments on four multimodal time series datasets, including Folsom, SKIPP'D, CCG, and CRNN. The input length $L$ is 48, and the prediction horizon $T$ is 24. The evaluation metrics are Mean Squared Error (MSE) and Mean Absolute Error (MAE), where lower values indicate better performance. Details for metrics are provided in App. B.5.

**Results.** The results are presented in Tab. 1. First, our proposed FACTS consistently outperforms all unimodal and multimodal baselines across the four datasets. For example, on Folsom, FACTS reduces MSE/MAE by 9.82%/13.18% compared with the second-best method. Similarly, on CRNN, FACTS surpasses the second-best method with reductions of 31.77%/17.48% in MSE/MAE. Furthermore, FACTS also exhibits smaller standard deviations than competing methods, which indicates greater stability. These significant gains demonstrate that incorporating auxiliary modalities effectively enhances TSF. Second, the teacher network, which accesses future auxiliary data, outperforms the student network, with average reductions of 11.98%/3.34% in MSE/MAE. These results suggest that future auxiliary data can provide precise causal drivers of temporal dynamics. By uncovering and transferring these cross-modal causal drivers via our CPCD, FACTS can effectively capture the underlying temporal dynamics, thereby delivering accurate and reliable predictions.

### 4.3 MODEL ANALYSIS

To evaluate the contribution of each component in our proposed FACTS, we conduct ablation studies on the Folsom and SKIPP'D datasets.

**Causal-Perturbation Contrastive Distillation (CPCD).** We replace CPCD with other KD methods, including widely used general-purpose methods of Feature Knowledge Distillation (FKD) (Zagoruyko & Komodakis, 2017) and Contrastive Representation Distillation (CRD) (Tian

et al., 2019); and SOTA distillation approaches TimeKD (Liu et al., 2025a) and TimeDistill (Ni et al., 2025b) tailored for TSF. Details of these KD methods are provided in App. B.4. As shown in the 'CPCD' row group of Tab. 2, FACTS equipped with CPCD achieves the best performance. When we replace our CPCD with TimeKD, which likewise leverages future information, the MSE/MAE worsen by 16.34%/23.45% on Folsom and by 10.84%/8.32% on SKIPP'D. These results indicate that CPCD can effectively uncover and transfer valuable causal signals from the future auxiliary data, enabling the student network to achieve reliable performance.

**Multimodal Data Fusion.** We compare the proposed lag-aware fusion with commonly used fusion techniques, including gating-based, self-attention–based, and channel-similarity-based methods (see App. C.7). The results are reported in the 'Multimodal Data Fusion' row group in Tab. 2. First, our lag-aware multimodal fusion, which fully accounts for inter-modal channel similarity and temporal lag, achieves the best predictive performance. Second, without considering the temporal lag (the 'Channel-Similarity' variant), the model's performance dropped significantly. The MSE/MAE increased by 13.26%/23.86% and 4.13%/2.62% on Folsom and SKIPP'D, respectively. These results indicate that it is critical to address cross-modality temporal misalignment during multimodal fusion.

**Backbone with Bidirectional Attention.** To evaluate the effectiveness of our proposed backbone, we benchmark it against classical TSF models (iTransformer (Liu et al., 2024c), TimeMixer (Wang et al., 2024b), and TimesNet (Wu et al., 2023)) and pre-trained LLMs (GPT-2 (Radford et al., 2019) and Llama-7B (Touvron et al., 2023)). Here, the classical TSF baselines are trained from scratch, like our backbone. In contrast, the original parameters of the LLMs are frozen, and they are updated via LoRA. The results are shown in 'Backbone' row group of Tab. 2. We can see that replacing our backbone with any of the alternatives leads to a significant performance drop. In the most severe case, MSE and MAE increased by 59.49% and 43.90% on Folsom, respectively. These results demonstrate that our backbone can facilitate accurate and reliable forecasts with its capability to capture both cross-variable interactions and cross-temporal dynamics.

**Bilinear Orthogonal Projector (BOP).** To validate the effectiveness of our BOP, we compare it with mainstream dimensionality-reduction methods, including Principal Component Analysis (PCA) (Abdi & Williams, 2010) and Independent Component Analysis (ICA) (Lee, 1998) (detailed in App. C.2). As shown in Fig. 2, our approach achieves superior performance with lower runtime compared with other methods. Moreover, when dropping BOP and directly applying an image encoder (VGGNet (Simonyan & Zisserman, 2014), ViT (Dosovitskiy et al., 2020), and CLIP (Radford et al., 2021), detailed in App. C.1) to process raw high-dimensional images, we observed a dramatic performance drop and a substantial computation increase. These re-

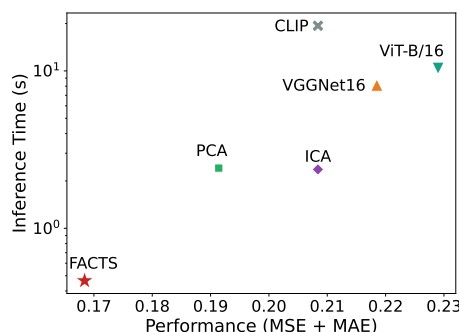

Figure 2: Performance (lower is better) on Folsom and per-multimodal-input inference latency of FACTS vs. BOP-replacement variants.

sults demonstrate that our proposed BOP can effectively convert images into time series, enabling efficient temporal information exploration.

### 4.4 HYPERPARAMETER ANALYSIS

In our FACTS, there are several tunable hyperparameters, including the trade-off weight $\lambda$ in objective function (Eq. (17)), the compression ratios $r_h$ and $r_w$ in BOP, and the window size $\delta_{\max}$ for lag-aware fusion. In this section, we analyze the sensitivity of FACTS to these parameters on Folsom and CRNN datasets, which involve two distinct application domains (solar power generation and water-level monitoring). During training, we vary one parameter while keeping the others fixed and record the corresponding results.

The curves of MSE and MAE under different hyperparameter settings are shown in Fig. 3. The parameter $\lambda$ balances the contributions of $\mathcal{L}_{\text{MSE}}$ and $\mathcal{L}_{\text{CPCD}}$, with values in $\{0.001, 0.01, 0.1, 1.0\}$. Even across a wide range, the MSE and MAE curves remain smooth and relatively stable, which

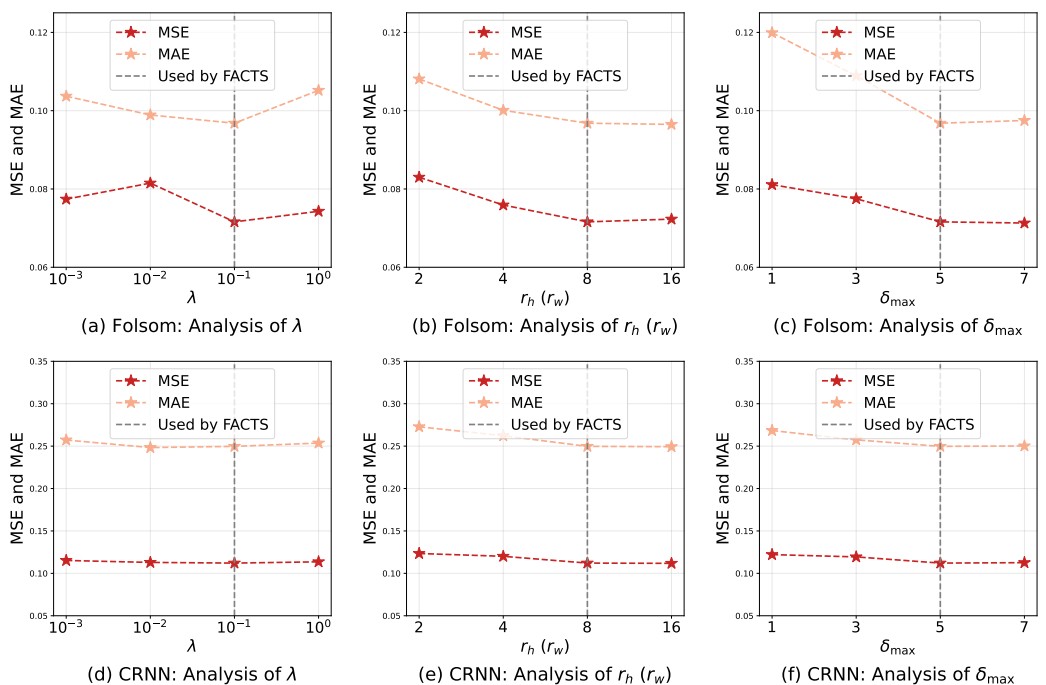

Figure 3: Analysis of $\lambda$ in objective function, $r_h$ ($r_h=r_w$) in bilinear orthogonal projector, and $\delta_{\max}$ for lag-aware fusion. *Note*: lower MSE/MAE indicates better model performance.

indicates that FACTS is robust to the variations of $\lambda$. Furthermore, FACTS consistently achieves competitive performance on both datasets when $\lambda = 0.1$, so we adopt this value for our experiments.

The parameters $r_h$ and $r_w$ determine the degree of data compression in BOP, with values in $\{2, 4, 8, 16\}$. To equally preserve spatial information along both height and width dimensions while maintaining BOP's parameter efficiency (see App. A.1), we set $r_h = r_w$ in our experiments. A small $r_h$ ($r_w$) implies strong compression and large information loss, resulting in reduced model performance. As $r_h$ ($r_w$) increases, more information is preserved and the model performance improves. When $r_h$ ($r_w$) reaches 8, the MSE and MAE curves begin to stabilize. Considering both efficiency and accuracy, we set $r_h = r_w = 8$ in our experiments.

The parameter $\delta_{\max}$ controls the lag window size for computing cross-modal similarity, with values in $\{1, 3, 5, 7\}$. Similar to the trend observed for $r_h$ ($r_w$), a small $\delta_{\max}$ insufficiently captures cross-modal temporal misalignment and results in inferior model performance. As $\delta_{\max}$ increases, the expanded lag window improves the model performance, and the curve stabilizes at $\delta_{\max} = 5$. To balance computational efficiency and predictive accuracy, we set $\delta_{\max} = 5$ in all experiments.

## 5 CONCLUSION

We introduced FACTS, a practical multimodal forecasting framework that (i) learns cross-modal causal responses with a future-aided teacher network and distills them to a deployable student network via CPCD, (ii) addresses modality heterogeneity with a BOP and a unified bidirectional-attention backbone spanning temporal and auxiliary inputs, and (iii) explicitly handles cross-modal misalignment via lag-aware fusion and modality missingness via random modality dropout. FACTS attains SOTA performance with improved stability on four real-world datasets, and ablations confirm that every component is necessary. To the best of our knowledge, FACTS is the first framework to jointly integrate temporal, image, and meteorological modalities for time series forecasting. By unifying heterogeneous modalities and distilling causal cues, FACTS offers an effective and efficient solution for realistic and challenging multimodal time series forecasting.

## REPRODUCIBILITY STATEMENT

We provide an anonymous code repository with training/evaluation scripts and configuration files to facilitate replication of all results. The main paper specifies the complete architecture and learning setup. Implementation details (optimizer, learning rate/weight decay, modality-dropout probability, loss weighting, lag window, BOP ranks), plus the "three runs with mean±std" reporting protocol, are documented in the Experimental Setup and hyperparameter analysis sections of the paper and appendix. We report ablations isolating each component in Sec. 4.3 and App. C, enabling verification of individual design choices. Together, these materials (code, sectioned descriptions, and appendix) are intended to make our results straightforward to reproduce and extend.

## ETHICS STATEMENT

This work studies multimodal time series forecasting using publicly available datasets on solar generation (with upward-facing sky cameras and meteorological measurements) and river water levels from the United States Geological Survey. We did not collect new data and did not process any data containing personally identifiable information or human subjects. No Institutional Review Board approval was required.

We adhered to dataset licenses and usage terms and will release code, configuration files, and documentation sufficient to reproduce the reported results, including data preprocessing steps and train/validation/test splits, to promote transparency and reproducibility. We took care to avoid temporal leakage across splits.

Potential risks include distribution shift and misuse of forecasts in safety-critical settings (*e.g.*, grid operations, flood response). Our method is intended for research and decision-support; it should not be deployed as the sole basis for real-time control without rigorous domain validation, uncertainty analysis, and human oversight. We discuss limitations and robustness (e.g., missing-modality sensitivity) and report results across multiple runs to mitigate over-claiming.

Fairness concerns may arise from geographical or climatological imbalances in the source datasets (*e.g.*, clear-sky prevalence, sensor coverage). We encourage future evaluations on diverse regions and conditions and provide implementation details to facilitate such auditing.

Regarding environmental impact, we report model sizes and training settings to enable estimation of computational cost. Our design includes efficient dimensionality-reduction components intended to reduce compute relative to high-resolution image processing baselines.

We believe this work complies with the ICLR Code of Ethics, including considerations of privacy, data governance, potential harms, fairness, and research integrity.

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

Table A-1: Inference costs and parameter footprints over $L$ square frames for our BOP and other compared methods.

| Method | Inference Costs (all $L$ frames) | Params |
|---|---|---|
| **PCA** | $\mathcal{O}(L\,S^2 K)$ | $\mathcal{O}(HWK)$ |
| **ICA** | $\mathcal{O}(L\,S^2 K)$ | $\mathcal{O}(HW\,K + K^2)$ |
| **BOP (ours)** | $\mathcal{O}\big(L\,(S^2\sqrt{K} + SK)\big)$ | $\mathcal{O}((H+W)\sqrt{K})$ |

# A  MODEL DETAILS

## A.1  INFERENCE COMPLEXITY AND PARAMETER FOOTPRINT OF DOWNSAMPLE METHODS

Given a high-dimensional image sequence $\mathbf{x}_{\text{img}} \in \mathbb{R}^{L \times H \times W}$, we analyze the end-to-end inference costs and parameter footprints for our BOP and two widely used frame-wise dimensionality-reduction methods, including Principal Component Analysis (PCA) and Independent Component Analysis (ICA).

**PCA.**  Firstly, each frame in $\mathbf{x}_{\text{img}}$ is vectorized as $\mathbf{x}_{\text{img},l}^{(\text{vec})} := \text{vec}(\mathbf{x}_{\text{img},l}) \in \mathbb{R}^{HW}$. Then, given the mean vector $\bar{\mathbf{x}}_{\text{img}}^{(\text{vec})} = \frac{1}{L}\sum_{l=1}^{L} \mathbf{x}_{\text{img},l}^{(\text{vec})}$ and the scatter matrix $\mathbf{G} = \frac{1}{L}\sum_{l=1}^{L} \big(\mathbf{x}_{\text{img},l}^{(\text{vec})} - \bar{\mathbf{x}}_{\text{img}}^{(\text{vec})}\big)\big(\mathbf{x}_{\text{img},l}^{(\text{vec})} - \bar{\mathbf{x}}_{\text{img}}^{(\text{vec})}\big)^{\top} \in \mathbb{R}^{HW \times HW}$, the projection matrix $\mathbf{W}_{\text{PCA}} \in \mathbb{R}^{HW \times K}$ is obtained by collecting the top-$K$ eigenvectors of $\mathbf{G}$ ($K \ll HW$). Finally, the PCA feature of frame $l$ is obtained by a single linear projection:

$$\mathbf{x}_{\text{img},l}^{(\text{PCA})} = \mathbf{W}_{\text{PCA}}^{\top}\big(\mathbf{x}_{\text{img},l}^{(\text{vec})} - \bar{\mathbf{x}}_{\text{img}}^{(\text{vec})}\big) \in \mathbb{R}^{K}, \qquad l = 1, \ldots, L. \tag{18}$$

*Inference complexity (all $L$ frames).* For each frame, mean subtraction costs $\mathcal{O}(HW)$ and the projection $\mathbf{W}_{\text{PCA}}^{\top}(\cdot)$ costs $\mathcal{O}(HWK)$. Hence, the end-to-end inference cost is

$$\text{cost}^{(\text{PCA})}(L) = \mathcal{O}(L\,H\,W\,K) = \mathcal{O}(L\,S^2\,K), \quad S = H = W. \tag{19}$$

*Parameter footprint.* Storing $\mathbf{W}_{\text{PCA}} \in \mathbb{R}^{HW \times K}$ requires $\mathcal{O}(HWK)$ parameters.

**ICA.**  Following the standard ICA pipeline, a (precomputed) *whitening* matrix $\mathbf{B} \in \mathbb{R}^{HW \times K}$ maps centered inputs to $K$-dimensional whitened features, and an ICA *demixing* (rotation) matrix $\mathbf{R} \in \mathbb{R}^{K \times K}$ enforces statistical independence:

$$\mathbf{W}_{\text{ICA}} := \mathbf{B}\,\mathbf{R}^{\top} \in \mathbb{R}^{HW \times K}.$$

The ICA feature of frame $l$ is

$$\mathbf{x}_{\text{img},l}^{(\text{ICA})} = \mathbf{W}_{\text{ICA}}^{\top}\big(\mathbf{x}_{\text{img},l}^{(\text{vec})} - \bar{\mathbf{x}}_{\text{img}}^{(\text{vec})}\big) \in \mathbb{R}^{K}, \qquad l = 1, \ldots, L. \tag{20}$$

*Inference complexity (all $L$ frames).* Per frame, mean subtraction costs $\mathcal{O}(HW)$ and the projection $\mathbf{W}_{\text{ICA}}^{\top}(\cdot)$ costs $\mathcal{O}(HWK)$. Hence, the end-to-end inference cost is

$$\text{cost}^{(\text{ICA})}(L) = \mathcal{O}(L\,H\,W\,K) = \mathcal{O}(L\,S^2\,K), \quad S = H = W. \tag{21}$$

*Parameter footprint.* Storing $\mathbf{W}_{\text{ICA}} \in \mathbb{R}^{HW \times K}$ requires $\mathcal{O}(HWK)$.

**BOP.**  For each frame $\mathbf{x}_{\text{img},l} \in \mathbb{R}^{H \times W}$, BOP applies a separable bilinear map with $\mathbf{U} \in \mathbb{R}^{H \times r_h}$ and $\mathbf{V} \in \mathbb{R}^{W \times r_w}$:

$$\mathbf{Y}_l = \mathbf{U}^{\top}\mathbf{x}_{\text{img},l}\mathbf{V} \in \mathbb{R}^{r_h \times r_w}, \qquad \mathbf{x}_{\text{img},l}^{(\text{BOP})} = \text{vec}(\mathbf{Y}_l) \in \mathbb{R}^{K}, \quad K := r_h r_w. \tag{22}$$

The orthogonality regularizers used during training incur no extra cost at inference.

*Inference complexity (all $L$ frames).* Per frame, evaluating $\mathbf{U}^{\top}\mathbf{x}_{\text{img},l}\mathbf{V}$ can be done in either order:

$$\underbrace{\mathcal{O}(HWr_h)}_{\mathbf{U}^{\top}\mathbf{x}} + \underbrace{\mathcal{O}(Wr_hr_w)}_{(\cdot)\mathbf{V}} = \mathcal{O}(HWr_h + WK),$$

or

$$\underbrace{\mathcal{O}(HWr_w)}_{\mathbf{xV}} + \underbrace{\mathcal{O}(Hr_hr_w)}_{\mathbf{U}^\top(\cdot)} = \mathcal{O}(HWr_w + HK).$$

Choosing the cheaper order yields

$$\text{cost}^{(\text{BOP})}(L) = \mathcal{O}\Big(L \cdot \min\{ HWr_h + WK, \ HWr_w + HK \}\Big). \tag{23}$$

In our BOP (square frames $H = W = S$) with balanced ranks $r_h = r_w = r$ (so $K = r^2$),

$$\text{cost}^{(\text{BOP})}(L) = \mathcal{O}\big(L\,(S^2 r + S r^2)\big) = \mathcal{O}\big(L\,(S^2\sqrt{K} + SK)\big). \tag{24}$$

*Parameter footprint.* BOP stores only the factor matrices:

$$\text{params}^{(\text{BOP})} = \mathcal{O}(Hr_h + Wr_w), \tag{25}$$

which under balanced ranks simplifies to $\mathcal{O}((H + W)\sqrt{K})$.

**Complexity and Parameter Ranking (lower is better).** Under square frames $H = W = S$ and balanced ranks for BOP ($r_h = r_w = \sqrt{K} \le S$), the per-sequence inference costs and parameter footprints satisfy

$$\boxed{\textbf{Inference complexity:} \quad \text{BOP} \ll \text{PCA} \approx \text{ICA}}$$

$$\boxed{\textbf{Parameter footprint:} \quad \text{BOP} \ll \text{PCA} \approx \text{ICA}}$$

These inequalities hold whenever $K \ll S^2$.

**Improved Parameter Efficiency via Balanced Rank.** Adopting a balanced-rank configuration ($r_h = r_w$) in BOP mathematically guarantees a minimized parameter footprint of $\mathcal{O}(Hr_h + Wr_w)$. For instance, when compressing a $64 \times 64$ image from Folsom dataset to an $8 \times 8$ representation, the balanced-rank setting ($r_h = r_w = 8$) requires only $1,024$ parameters ($64 \times 8 + 64 \times 8$). In comparison, an unbalanced configuration of equivalent capacity (*e.g.*, $r_h = 2, r_w = 32$) necessitates $2,176$ parameters ($64 \times 2 + 64 \times 32$), demonstrating the significant efficiency gains of the balanced approach.

A.2 DETAILS OF BIDIRECTIONAL-ATTENTION BACKBONE.

**Time Series Partitioning and Embedding** Given the input series $\mathbf{x} \in \mathbb{R}^{L \times V}$, we first partition it along the temporal axis into $P$ patches of length $S$ with stride $r$:

$$\mathbf{x}_{(p)} = \mathbf{x}_{t_p:t_p+S-1,:} \in \mathbb{R}^{S \times V}, \quad t_p = 1 + (p-1)r, \quad P = \left\lfloor \frac{L - S}{r} \right\rfloor + 1. \tag{26}$$

Here, $p \in \{1, \cdots, P\}$, we omit modality-specific superscripts and subscripts for notational simplicity. Then, each patch is linearly projected along the temporal dimension to produce a $D$-dimensional embedding per variable:

$$\mathbf{z}_p = \mathbf{x}_{(p)}^\top \mathbf{W}_t + \mathbf{b}, \quad \mathbf{W}_{\text{proj}} \in \mathbb{R}^{S \times D}, \mathbf{b}_{\text{proj}} \in \mathbb{R}^D. \tag{27}$$

Here, $\mathbf{W}_{\text{proj}}$ and $\mathbf{b}_{\text{proj}}$ are the weight and bias of the projection layer, respectively. The temporal embeddings for $P$ patches are stacked as the patching embedding $\mathbf{z} \in \mathbb{R}^{P \times V \times D}$.

**Forecaster.** Given the encoder output $\mathbf{h} \in \mathbb{R}^{P \times V \times D}$, we first flatten each variable to obtain:

$$\mathbf{h}_{\text{reshape}} = \text{reshape}(\mathbf{h}, \ V \times (PD)) \in \mathbb{R}^{V \times (PD)}. \tag{28}$$

The forecaster $g_\phi$ is a two-layer MLP that maps $\mathbf{H}$ to a $T$-step forecast:

$$\mathbf{h}_{\text{reshape}}^{(1)} = \sigma\Big(\mathbf{h}_{\text{reshape}} \mathbf{W}_\phi^{(1)} + \mathbf{1}\, b_\phi^{(1)\top}\Big) \in \mathbb{R}^{V \times d_f}, \tag{29}$$

$$\mathbf{h}_{\text{reshape}}^{(2)} = \mathbf{h}_{\text{reshape}}^{(1)} \mathbf{W}_\phi^{(2)} + \mathbf{1}\, b_\phi^{(2)\top} \in \mathbb{R}^{V \times T}, \tag{30}$$

$$\hat{\mathbf{y}} = \mathbf{h}_{\text{reshape}}^{(2)\top} \in \mathbb{R}^{T \times V}. \tag{31}$$

Here, $\mathbf{W}_\phi^{(1)} \in \mathbb{R}^{(PD) \times d_f}$, $\mathbf{b}_\phi^{(1)} \in \mathbb{R}^{d_f}$, $\mathbf{W}_\phi^{(2)} \in \mathbb{R}^{d_f \times T}$, $\mathbf{b}_\phi^{(2)} \in \mathbb{R}^T$, $\sigma(\cdot)$ denotes a nonlinearity (*e.g.*, GELU), $d_f$ is the hidden size, and $\mathbf{1}$ is an all-ones column vector for bias broadcasting.

Table A-2: Model configurations of our FACTS.

| Parameter | Value | Description |
|---|---|---|
| $P$ | 4 | Number of patches of the segmented input sequence. |
| $S$ | 12 | Patch length for each segment. |
| $r$ | 12 | Stride for patching, equals $S$ for non-overlapping patches. |
| $D$ | 512 | Patch embedding dimension. |
| $N_{\text{BA}}$ | 4 | Number of bidirectional-attention blocks. |
| $n_{\text{heads}}$ | 4 | Number of attention heads. |
| Enc_in | $V_{\text{img}}, V_{\text{wea}}$ or $V_{\text{time}}$ | Number of input channels (variables). |
| Enc_out | $V_{\text{time}}$ | Number of output channels. |

Table A-3: The details of multimodal benchmark datasets.

| Field | Dataset | Variate | Frequency | Time Range | Modality |
|---|---|---|---|---|---|
| Solar Power | Folsom | 42 | 5 mins | 2014.01-2016.12 | Temporal, Image, Weather |
| Generation | SKIPP'D | 1 | 2 mins | 2017.03-2019.10 | Temporal, Image |
| Water-Level | CCG | 1 | 15 mins | 2024.01-2025.07 | Temporal, Image |
| Monitoring | CRNN | 1 | 1 hour | 2023.12-2024.08 | Temporal, Image |

### A.3 BRANCH-SPECIFIC MODEL CONFIGURATION

To account for the heterogeneity in variable dimensionality across modalities, we employ branch-specific parameterization for each modality branch. Detailed configurations are provided in Tab. A-2. Except for the number of input channels (which equals the number of variables in each modality), all branches share identical parameter configurations. Therefore, our multimodal multi-branch architecture does not introduce additional parameter-configuration overhead.

### A.4 TEACHER TRAINING AND MODEL SETTINGS

**Training of Teacher Network.** Given the time series $\mathbf{x}_{\text{time}}^i$ and the concatenated auxiliary sequences $\mathbf{z}_{\text{img}}^i$ and $\mathbf{z}_{\text{wea}}^i$, we first input them into the teacher network and obtain the modality-specific predictions as follows:

$$\hat{\mathbf{y}}_{\text{time}}^{\text{Tea}} = g_{\phi_{\text{time}}}^{\text{Tea}}(f_{\theta_{\text{time}}}^{\text{Tea}}(\mathbf{x}_{\text{time}})), \quad \hat{\mathbf{y}}_{\text{img}}^{\text{Tea}} = g_{\phi_{\text{img}}}^{\text{Tea}}(f_{\theta_{\text{img}}}^{\text{Tea}}(\mathcal{B}^{\text{Tea}}(\mathbf{z}_{\text{img}}))), \quad \hat{\mathbf{y}}_{\text{wea}}^{\text{Tea}} = g_{\phi_{\text{wea}}}^{\text{Tea}}(f_{\theta_{\text{wea}}}^{\text{Tea}}(\mathbf{z}_{\text{wea}})). \quad (32)$$

Here, the superscript 'Tea' is employed to denote teacher network, distinguishing it from the prediction horizon $T$. Then, we fuse the modality-specific predictions based on their similarity scores to obtain the final forecast $\hat{\mathbf{y}}_{\text{Tea}}^i \in \mathbb{R}^{T \times V}$, following the process detailed in Sec. 3.2. The teacher is optimized with the mean squared error (MSE):

$$\mathcal{L}_{\text{MSE}}^{\text{Tea}} = \frac{1}{BTV_{\text{time}}} \sum_{i=1}^{B} \sum_{t=1}^{T} \sum_{v=1}^{V_{\text{time}}} (\mathbf{y}_{t,v}^i - \hat{\mathbf{y}}_{\text{Tea},t,v}^i)^2, \quad (33)$$

where $B$, $T$, and $V_{\text{time}}$ denote the batch size, prediction horizon, and number of variables, respectively. By minimizing $\mathcal{L}_{\text{MSE}}^{\text{Tea}}$, the teacher network is promoted to predict as accurately as the ground-truth series $\mathbf{y}$.

**Model Configurations.** Because the teacher network ingests both historical and future auxiliary data, the input length of its auxiliary branches is $L + T$, whereas the student network uses only historical auxiliary data with length $L$. Apart from this difference in input length, the two networks share the same model configuration, as shown in Tab. A-2.

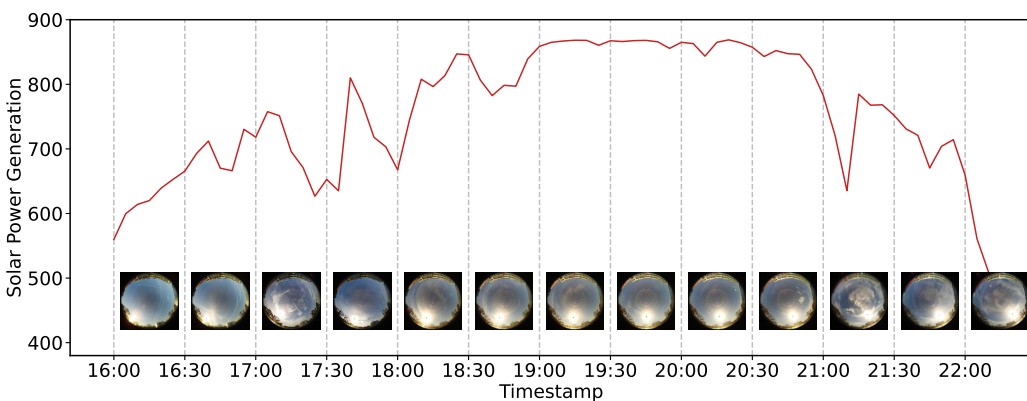

Figure A-1: Visualization of the time series and the corresponding sky images in the Folsom dataset. Here, the dashed interval displays the earliest image in that window. For example, the 17:00–17:30 interval shows the photo taken at 17:00. The sky image at 17:00 shows extensive cloud occlusion. However, the solar power output continues to rise for the next several minutes before dropping sharply. This indicates a temporal mismatch (lag) between the image and temporal modalities.

## B  ADDITIONAL EXPERIMENTAL SETUPS

### B.1  BENCHMARK DATASETS

In this paper, we evaluate our proposed FACTS on four publicly available multimodal time series datasets, including two solar power generation datasets (Folsom and SKIPP'D) and two water-level monitoring datasets (CCG and CRNN). Detailed descriptions (summarized in Tab. A-3) of the datasets are provided below:

**Folsom** comprises three consecutive years (2014.01–2016.12) of solar-irradiance measurements collected in Folsom, California, which are directly related to photovoltaic power generation. It includes 5-minute–resolution ground irradiance (the target time series for forecasting), one all-sky camera image per minute, and meteorological observations. The time series component provides the same set of seven irradiance-related variables at six lead times (5, 10, 15, 20, 25, and 30 minutes), yielding 42 variables in total. The seven variables are Global Horizontal Irradiance (GHI), Direct Normal Irradiance (DNI), clear-sky GHI, clear-sky DNI, Clear-sky–normalized GHI, Clear-sky–normalized DNI, and solar elevation angle. The meteorological observations contain seven variables, including air temp, relative humidity, pressure, wind speed, wind direction, and precipitation.

**SKIPP'D** is a SKy Images and Photovoltaic Power Generation Dataset for short-term solar forecasting, which is collected on Stanford University's campus. It comprises three consecutive years (2017.03–2019.10) of synchronized data, including a ready-to-use benchmark with 1-minute resolution, down-sampled all-sky images (64×64) paired with minutely averaged solar power series (with 1 variable).

**Clear Creek in Golden (CCG)** is a water-level monitoring dataset from the official website of the United States Geological Survey (USGS) [1]. The monitoring site is located at Clear Creek in Golden, and the time span is from 2024.01 to 2025.07. The benchmark uses 15-minute resolution and pairs downsampled river-scene images (64×64) with a single water-level time series sampled every 15 minutes, targeting short-horizon stream-level forecasting.

**Connecticut River Near Northfield (CRNN)** is also a water-level monitoring dataset from USGS, covering from 2023.12 to 2024.08. It adopts the 1-hour resolution, which contains 64×64 river images synchronized with one water-level variable. CRNN and CCG are collected from different sites, and thus representing distinct flow dynamics and seasonality.

---

[1] https://waterdata.usgs.gov

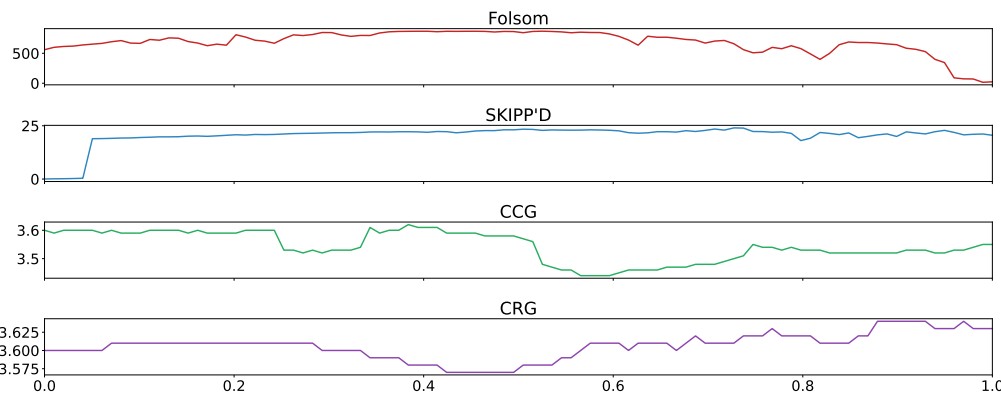

Figure A-2: Visualization results of the time series data from the four benchmark datasets. The time series exhibits distinct temporal patterns across datasets.

As illustrated in Fig. A-2, these datasets from various fields are collected from distinct locations and exhibit different temporal patterns. Despite these challenges, FACTS performs consistently well across multiple datasets, indicating strong generalization.

### B.2 VISUALIZATION OF TEMPORAL LAGS

As discussed in Sec. 3.2, cross-modal signals often exhibit temporal misalignment. Fig. A-1 shows that in the image for the 17:00–17:30 interval (captured at 17:00), the sun is obscured by clouds; however, the power output does not drop immediately at 17:00 but rises briefly before falling sharply. This observation indicates that temporal misalignment indeed occurs in practice. To address this issue, we propose a lag-aware multimodal fusion mechanism.

### B.3 BASELINE TIME SERIES FORECASTING METHODS

In this paper, we compare an extensive range of SOTA Time Series Forecasting (TSF) methods, primarily categorized as follows:

(i) **Transformer-based Unimodal Methods**:

- Crossformer (Zhang & Yan, 2023) identifies that the crossvariable relationships in time series data are crucial for TSF and captures them using attention mechanisms.
- FEDformer (Zhou et al., 2022) and Autoformer (Wu et al., 2021), which decouple seasonal and trend components in the frequency domain and learn them based on the attention mechanism.
- PatchTST (Nie et al., 2023c), the first work proposed partitioning input series into multiple patches, effectively enhancing the long-range TSF capability of Transformers.
- iTransformer (Liu et al., 2024c) transposes the input time series and implements the attention mechanism along the variable dimension to capture relationships between variables.
- ETSformer (Woo et al., 2022) introduces both smoothing attention and frequency attention to replace the original self-attention mechanism in Transformers, which can effectively extract the temporal patterns in input series.

(ii) **CNN-Based Unimodal Methods**:

- TimesNet (Wu et al., 2023), which selects representative periods in the frequency domain to construct an image and processes such an image using 2D convolution layers.
- TCN (Bai et al., 2018) conducts a systematic evaluation of generic convolutional and recurrent architectures for sequence modeling.
- MICN (Wang et al., 2022) decomposes the time series signal into seasonal and trend components and learns them separately using convolutional and linear regression layers.

(ii) **MLP-Based Unimodal Methods**:

- DLinear (Zeng et al., 2023) explores the application of linear layers in time series tasks and achieves efficient time series prediction.
- TiDE (Das et al., 2023) designs an encoder-decoder structure based on MLP, which can achieve comparable performance with Transformers while requiring fewer computations.
- TimeMixer (Wang et al., 2024b) downsamples the time series into multiple-scale inputs for ensemble predictions in the MLP model.

(iv) **LLM-Based Unimodal Methods**:

- GPT4TS (Zhou et al., 2023), the pioneering work that employs LLM for TSF by segmenting continuous time series into discrete tokens compatible with LLM.
- TimeLLM (Jin et al., 2024), which proposes patch reprogramming to encode prior knowledge from time series datasets into prompts for guiding the LLM in TSF.
- CALF (Liu et al., 2024b) trains separate branches for temporal and textual modalities and closely aligns them with leveraging textual knowledge in LLMs for time series prediction.

(v) **Multimodal Methods**:

- TimeVLM (Zhong et al., 2025) converts the input time series into a textual description and a spectrum image, and processes them with a pre-trained VLM. Then, the outputs from various modality branches are fused together as the final prediction.
- AimTS (Chen et al., 2025) transfers time series into a line chart and aligns temporal feature and image feature via contrastive learning.

## B.4 KNOWLEDGE DISTILLATION BASELINES

To verify the effectiveness of our proposed causal-perturbation contrastive distillation, we conduct comparative experiments against **classical knowledge distillation** methods and existing **time series distillation** techniques. Classical knowledge distillation has been widely applied in image recognition, natural language processing, and other domains, effectively extracting and transferring knowledge to improve model performance. The main approaches include:

- Feature Knowledge Distillation (FKD) (Zagoruyko & Komodakis, 2017) suggests intermediate features contain rich knowledge and performs distillation by aligning teacher and student features at intermediate and penultimate layers.
- Contrastive Representation Distillation (CRD) (Tian et al., 2019) brings paired teacher–student features closer in the representation space while pushing apart non-paired features, thereby improving the representational ability of the student network.

In recent years, knowledge distillation has also gained traction in time series forecasting, which is employed to transfer temporal knowledge from powerful pre-trained models to enhance the predictive performance of target models. Representative methods include:

- TimeDistill (Ni et al., 2025b) distills multi-scale and multi-period temporal signals from complex pre-trained models (*e.g.*, Transformers) into a simple MLP, successfully promoting the simple MLP to equip comparable performance with those complicated ones.
- TimeKD (Liu et al., 2025a) utilizes a model with access to future series as the teacher network to learn high-quality temporal representations and transfer them to a student network that observes only historical inputs.

## B.5 METRICS OF TIME SERIES FORECASTING

In this paper, we mainly employ four widely used metrics to assess model performance, including Mean Squared Error (MSE) and Mean Absolute Error (MAE).

**MSE** measures the average of the squared differences between the predicted and actual values. MSE gives more weight to more significant errors because the errors are squared, making it sensitive to

Table A-4: Parameter counts (in millions, M), Floating-Point Operations (FLOPs) (in gigas, G), and forecasting errors for alternative image branches and our FACTS image branch. *Note:* all STD values in the table are scaled by $\times 10^{-2}$.

| Image Branch | Parameters (M) | FLOPs (G) | MSE | STD | MAE | STD |
|---|---|---|---|---|---|---|
| **VGGNet16** | 14.72 | 1.26 | 0.0901 | 0.16 | 0.1284 | 0.37 |
| **ViT-B/16** | 85.65 | 16.86 | 0.0981 | 0.24 | 0.1301 | 0.44 |
| **CLIP** | 427.60 | 116.29 | 0.0891 | 0.20 | 0.1193 | 0.22 |
| **FACTS (Ours)** | 2.36 | 0.26 | **0.0716** | 0.03 | **0.0968** | 0.05 |

outliers. Given $T$ steps ground-truth time series signal $\mathbf{y} \in \mathbb{R}^{T \times V_{\text{time}}}$ and prediction $\hat{\mathbf{y}} \in \mathbb{R}^{T \times V_{\text{time}}}$, MSE is calculated as:

$$\text{MSE} = \frac{1}{TV_{\text{time}}} \sum_{t=1}^{T} \sum_{v=1}^{V_{\text{time}}} (\mathbf{y}_{t,v} - \hat{\mathbf{y}}_{t,v})^2 \, . \tag{34}$$

**MAE** quantifies the average absolute differences between predicted and actual values. It is less sensitive to outliers than MSE because it does not square the errors, treating all errors linearly. MAE is computed as:

$$\text{MAE} = \frac{1}{TV_{\text{time}}} \sum_{t=1}^{T} \sum_{v=1}^{V_{\text{time}}} |\mathbf{y}_{t,v} - \hat{\mathbf{y}}_{t,v}| \, . \tag{35}$$

*Note:* Both metrics are 'lower is better'.

## C MODEL ANALYSIS

### C.1 OVERVIEW OF THE ALTERNATIVE IMAGE ENCODERS

To evaluate the effectiveness of our proposed bilinear orthogonal projector, we remove it and conduct controlled experiments. After removal, the input dimensionality of the image branch changes from $\mathbb{R}^{V_{\text{img}}}$ ($V_{\text{img}} := r_h r_w$, $r_h \ll H$ and $r_w \ll W$) to $\mathbb{R}^{H \times W}$, which prevents it from processing high-dimensional raw images. Accordingly, we replace the original image branch with either trained-from-scratch or pretrained image models. These alternatives take high-dimensional images $\mathbf{x}_{\text{img}} \in \mathbb{R}^{H \times W}$ as input and output a time series $\hat{\mathbf{y}}_{\text{img}} \in \mathbb{R}^{T \times V_{\text{time}}}$.

**Trained-From-Scratch Image Models.** We adopt the classic VGGNet16 (Simonyan & Zisserman, 2014) and ViT-B/16 (Dosovitskiy et al., 2020) as replacement image branches and substitute their classifier with a temporal projector, which projects the image feature as the time series.

**Pretrained Image Models.** Compared with image models trained from scratch, pretrained models learn stronger representations from large-scale image datasets. Therefore, we use the CLIP's (Radford et al., 2021) image encoder as replacement image branches. Tab. A-4 reports the computational and parameter costs of our image branch and the alternatives. We can observe that our image branch has substantially fewer parameters and requires far less computation, while FACTS achieves the best forecasting performance. These results indicate that our image branch with the bilinear orthogonal projector can efficiently and effectively extract meaningful temporal signals from images to improve time series forecasting.

### C.2 IMAGE DIMENSIONALITY REDUCTION METHODS

To validate the effectiveness of our bilinear orthogonal projector, we replace it with alternative dimensionality-reduction techniques and conduct controlled experiments, including:

- Principal Component Analysis (PCA) (Abdi & Williams, 2010). Input images are first flattened and mean-centered to learn principal directions via covariance decomposition. Then, each image is projected to the top $K$ principal components as a low-dimensional vector.

Table A-6: Computational cost and performance comparison on Folsom dataset.

| Model | FLOPs (Giga) | Training Time (H) | MSE | STD | MAE | STD |
|---|---|---|---|---|---|---|
| **AimTS** | 5.34 | 3.65 | 0.1366 | 1.26 | 0.1843 | 1.62 |
| **TimeVLM** | 31.03 | 9.7 | 0.1210 | 0.15 | 0.1599 | 0.24 |
| **FACTS (Proposed)** | 0.93 | 1.54 | **0.0716** | **0.03** | **0.0968** | **0.05** |

Table A-7: Parameter sensitivity analysis results of $\alpha$ in random modality dropout.

| Missing Modality | $\alpha$=0.0 | | $\alpha$=0.2 | | $\alpha$=0.4 | | $\alpha$=0.8 | |
|---|---|---|---|---|---|---|---|---|
| | MSE | MAE | MSE | MAE | MSE | MAE | MSE | MAE |
| **Image** | 0.1729 | 0.2883 | 0.0916 | 0.1589 | 0.0842 | 0.1304 | 0.0964 | 0.1542 |
| **Weather** | 0.0958 | 0.2232 | 0.0924 | 0.1683 | 0.0781 | 0.1193 | 0.0933 | 0.1430 |
| **Image and Weather** | 0.0353 | 0.4359 | 0.1146 | 0.1685 | 0.0937 | 0.1457 | 0.1169 | 0.1662 |
| **N/A** | 0.0713 | 0.0961 | 0.0711 | 0.0973 | 0.0716 | 0.0968 | 0.0873 | 0.1351 |

- Independent Component Analysis (ICA) (Lee, 1998). Flattened images are first mean-centered and whitened, after which a set of statistically independent bases is learned by maximizing non-Gaussianity or information. Independent components under these bases represent each image, and the top $K$ components are used as reduced features.

## C.3 RANDOM MODALITY DROPOUT

To enhance the robustness of our method against modality missingness, which is common in real-world applications, we adopt a random modality dropout strategy during the training of student networks. To evaluate its effectiveness, we drop one or more modalities and estimate the performance of the trained student network. As reported in Tab. A-5, when auxiliary modalities are missing, the student network trained without random modality dropout

Table A-5: Results of student networks with missing modalities. *Note:* all STD values in the table are scaled by $\times 10^{-2}$.

| Missing Modality | Algorithm | MSE | STD | MAE | STD |
|---|---|---|---|---|---|
| Image | w/o RMD | 0.1729 | 1.08 | 0.2883 | 1.75 |
| | with RMD | 0.0842 | 0.27 | 0.1304 | 0.59 |
| Weather | w/o RMD | 0.0958 | 0.25 | 0.2232 | 0.73 |
| | with RMD | 0.0781 | 0.09 | 0.1193 | 0.18 |
| Image and Weather | w/o RMD | 0.3553 | 1.96 | 0.4359 | 2.80 |
| | with RMD | 0.0937 | 0.32 | 0.1457 | 0.53 |
| N/A | FACTS | **0.0716** | 0.03 | **0.0968** | 0.05 |

suffers a severe performance degradation. In comparison, the student network trained with random modality dropout experiences only minor performance fluctuations under missing modalities. These results demonstrate that our random modality dropout strategy can effectively improve the model's robustness in modality missingness scenarios.

We also analyze the dropout ratio $\alpha$ in our employed random modality dropout, the results are reported in Tab. A-7. We can observe that when $\alpha$=0.0 (no modalities are dropped during training), the model's performance degrades significantly in the presence of missing modalities. When $\alpha$=0.4, the model suffers only minor performance losses under modality absence, so we set $\alpha$=0.4 in our experiments. When $\alpha$=0.8, a large fraction of modalities are dropped, which severely disrupts the student training and leads to an obvious decline in performance.

## C.4 COMPUTATIONAL COST ANALYSIS

Our FACTS adopts a teacher-student architecture. The teacher network is used only during training to distill causal knowledge into the student network, while the student network serves as the final deployable model. Here, we provide a detailed analysis of the computational overheads incurred by FACTS and other multimodal methods. The FLoating-point OPerations (FLOPs) and training time (in hours, H) of all methods are reported in Tab. A-6. All experiments are conducted on Folsom dataset using an NVIDIA RTX 4090 GPU. Even though FACTS includes both teacher and student

networks, it requires only 0.93G FLOPs, which is significantly lower than AimTS (5.34G) and TimeVLM (31.03G). This indicates that our FACTS is highly efficient for real-world inference.

## C.5 Underlying Data Assumptions

Our FACTS aims to leverage auxiliary modalities through a teacher network during training to uncover causal drivers, and then transfer this knowledge to a deployable student network that relies solely on historical data, thereby enhancing the performance and reliability of student network. To achieve this, there are two key and readily satisfied underlying data assumptions:

1. Future auxiliary data (*e.g.*, sky images) contains true causal drivers that reveal temporal dynamics. This condition is validated by the results in Tab. 1, where teacher network (with access to future auxiliary data) consistently outperforms student network (which relies solely on historical data). This demonstrates that the future auxiliary data indeed carries essential causal information.

2. The perturbed examples (with random future data) can be clearly distinguished from the unperturbed examples (with true future data). This condition is supported by the visualization results in Fig. A-3, which show a clear separation between the unperturbed (red) and perturbed (green) features.

## C.6 T-SNE Visualization for CPCD

To intuitively validate the effectiveness of our CPCD, we conducted a t-SNE visualization on the test set of Folsom dataset. As shown in Fig. A-3, the teacher network's unperturbed features (red, representing 'faithful causal knowledge') and perturbed features (green, representing 'spurious correlations') correspond to two clearly separated trajectories. This suggests that the perturbation introduces substantial shifts in the feature space of teacher network, enabling CPCD to construct meaningful contrasting signals.

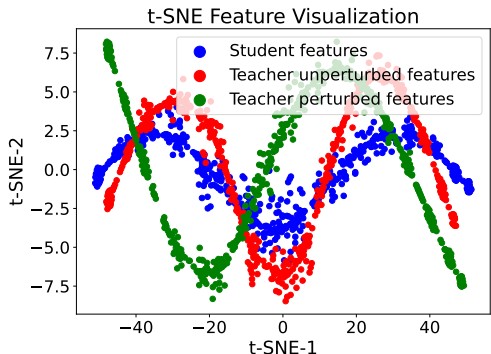

Figure A-3: t-SNE visualization on the test set of Folsom dataset.

More importantly, the features of student network (blue) largely follow the same manifold as the unperturbed features of teacher network (red), while remaining clearly separated from the perturbed features of teacher network (green). This visualization indicates that CPCD successfully guides student network to learn the representations aligned with the 'faithful causal knowledge' extracted by teacher network, while repelling them from the 'spurious correlations', and thus achieving effective causal disentanglement.

## C.7 Compared Multimodal Data Fusion Methods

To verify the effectiveness of our lag-aware multimodal fusion mechanism, we compare it against three fusion methods, including gating-based, attention-based, and simple-similarity-based approaches. For completeness, we also report a simple similarity-based fusion baseline that ignores temporal misalignment.

**Gating-Based Method.** Given future series $\hat{\mathbf{y}}_{\text{time}}, \hat{\mathbf{y}}_{\text{img}}, \hat{\mathbf{y}}_{\text{wea}} \in \mathbb{R}^{T \times V_{\text{time}}}$ predicted by various modality branches, a gating network is employed to map them to gating tensors $\mathbf{g}_{\text{time}}, \mathbf{g}_{\text{img}}, \mathbf{g}_{\text{wea}} \in [0,1]^{T \times V_{\text{time}}}$. Then, these future series are fused as:

$$\hat{\mathbf{y}} = \hat{\mathbf{y}}_{\text{time}} \odot \mathbf{g}_{\text{time}} + \hat{\mathbf{y}}_{\text{img}} \odot \mathbf{g}_{\text{img}} + \hat{\mathbf{y}}_{\text{wea}} \odot \mathbf{g}_{\text{wea}}. \tag{36}$$

**Attention-Based Method.** These modality-specific sequences are concatenated and fused by a Multi-Head Self-Attention (MHSA) layer, as follows:

$$\hat{\mathbf{y}} = \text{MHSA}_{\text{fuse}}(\hat{\mathbf{y}}_{\text{time}}, \hat{\mathbf{y}}_{\text{img}}, \hat{\mathbf{y}}_{\text{wea}}). \tag{37}$$

**Simple-Similarity-Based Method.** As a simple baseline, it directly computes channel-wise similarities among $\hat{\mathbf{y}}_{\text{time}}$, $\hat{\mathbf{y}}_{\text{img}}$, and $\hat{\mathbf{y}}_{\text{wea}}$ to derive static weights, as follows:

$$w_{\text{img}}^{(v)} = \sum_{t=1}^{T} \hat{\mathbf{y}}_{\text{time}, t, v} \, \hat{\mathbf{y}}_{\text{img}, t, v}, \quad w_{\text{wea}}^{(v)} = \sum_{t=1}^{T} \hat{\mathbf{y}}_{\text{time}, t, v} \, \hat{\mathbf{y}}_{\text{wea}, t, v}, \quad v \in \{1, \cdots V_{\text{Time}}\}. \tag{38}$$

Then, given the similarities $\mathbf{w}_{\text{img}} = \left[ w_{\text{img}}^{(1)}, \ldots, w_{\text{img}}^{(V_{\text{time}})} \right]^{\top} \in \mathbb{R}^{V_{\text{time}}}$ and $\mathbf{w}_{\text{wea}} = \left[ w_{\text{wea}}^{(1)}, \ldots, w_{\text{wea}}^{(V_{\text{time}})} \right]^{\top} \in \mathbb{R}^{V_{\text{time}}}$, the final prediction $\hat{\mathbf{y}}$ are obtained as:

$$\hat{\mathbf{y}} = \hat{\mathbf{y}}_{\text{time}} + \hat{\mathbf{y}}_{\text{img}} \odot \mathbf{w}_{\text{img}} + \hat{\mathbf{y}}_{\text{wea}} \odot \mathbf{w}_{\text{wea}}. \tag{39}$$

## STATEMENT ON LLM USAGE

We used a large language model (LLM; *e.g.*, ChatGPT) solely as an editorial aid to polish the manuscript's prose. The LLM was *not* used for research ideation, model or algorithm design, dataset curation, experiment setup, code writing, analysis, or result generation. All technical contributions, experiments, figures/tables, and conclusions were conceived and produced by the authors, and all LLM-suggested edits were manually reviewed for accuracy and originality; citations were inserted and verified by the authors. No non-public data was shared with the LLM.

