# OpenReview forum: "FACTS: A Future-Aided Causal Teacher-Student Framework for Multimodal Time Series Forecasting"
_ICLR.cc/2026/Conference — Submitted to ICLR 2026_

### Official Review · Reviewer_UwUE · 2025-10-29

**Soundness:** 3
**Presentation:** 3
**Contribution:** 3
**Rating:** 6
**Confidence:** 4

**Summary:**

This paper proposes FACTS, a Future-Aided Causal Teacher-Student framework for multimodal time series forecasting. The key idea is to leverage future auxiliary modalities (such as images and weather data) during training to uncover causal factors that drive temporal dynamics, and distill this knowledge into a student network that uses only historical data at inference.

The method introduces three main components: (1) a Bilinear Orthogonal Projector (BOP) that converts high-dimensional auxiliary data into compact serialized time series; (2) a lag-aware multimodal fusion mechanism with random modality dropout to handle temporal misalignment and missing modalities; and (3) a Causal-Perturbation Contrastive Distillation (CPCD) objective that transfers causal knowledge from the teacher to the student. Experiments on four multimodal forecasting datasets show significant performance gains (around 33% lower MSE on average) over state-of-the-art baselines.

**Strengths:**

1. The proposed teacher-student design, which leverages future multimodal signals to facilitate causal representation learning while maintaining a purely historical input during inference, is both conceptually elegant and practically impactful. It effectively balances realism (no future data at test time) with enhanced learning through privileged information.

2. The model is comprehensively evaluated on several diverse and representative datasets, demonstrating the framework’s robustness and adaptability across various multimodal forecasting scenarios.

3. The experiments include thorough comparisons against a wide range of baselines, covering both unimodal and multimodal forecasting methods, thereby providing convincing evidence of the model’s superior performance and general applicability.

4. The paper conducts detailed ablation studies that isolate and analyze the contribution of each component—such as BOP, CPCD, and lag-aware fusion—clearly demonstrating their necessity and complementary effects within the overall framework.

**Weaknesses:**

see in questions.

**Questions:**

1. The paper does not clearly specify the training schedule of the teacher–student framework. Is the teacher network fully trained to convergence before the student distillation begins, or are the two networks optimized jointly in an alternating or end-to-end manner?

2. The Bilinear Orthogonal Projector (BOP) is primarily demonstrated on image data. Could the authors elaborate on its generality—specifically, whether BOP can be effectively adapted to other high-dimensional modalities such as text embeddings, audio spectrograms, or video streams? If not, what limitations might arise when extending it beyond visual inputs?

3. How significant is the computational overhead incurred by using both teacher and student networks, especially when compared to alternative approaches? Could the authors provide quantitative estimates or benchmarks to clarify this aspect?

4. The paper claims that random modality dropout enhances robustness when certain modalities are missing at inference time. Have the authors conducted experiments under more extreme or persistent missing-modality scenarios (e.g., complete loss of image data throughout inference)? Such evaluation would provide stronger evidence for the model’s robustness and practical deployability in real-world settings with unreliable sensors.

5. Is the ratio 𝜆 between the MSE loss and the CPCD loss a manually set hyperparameter or a learnable parameter? What is the value of this parameter, and how does varying 𝜆 affect the final model performance?

---

> ### Author Response · Authors · 2025-11-19
> **Response to Reviewer UwUE, part 1**
>
> Q1: The paper does not clearly specify the training schedule of the teacher–student framework. Is the teacher network fully trained to convergence before the student distillation begins, or are the two networks optimized jointly in an alternating or end-to-end manner?
>
> RQ1: Thank you for raising this critical clarification question. Our FACTS **trains teacher network to convergence before student network distillation**. This training scheme ensures that teacher network acquires sufficiently strong representational capacity to extract reliable causal signals for student network. To avoid ambiguity, we have clarified the teacher–student training procedure at the end of **Section 3.3** in the revised manuscript.
>
> Q2: The Bilinear Orthogonal Projector (BOP) is primarily demonstrated on image data. Could the authors elaborate on its generality—specifically, whether BOP can be effectively adapted to other high-dimensional modalities such as text embeddings, audio spectrograms, or video streams? If not, what limitations might arise when extending it beyond visual inputs?
>
> RQ2: This is a thought-provoking question. BOP is a frame-level bilinear orthogonal projection that is inherently suitable for 2D data (*e.g.*, images, spectrograms, or audio spectrograms). For such data, it can efficiently learn spatial projections by leveraging its bilinear structure. For 1D data (*e.g.*, texts), however, standard linear projections (*e.g.*, MLPs) are more straightforward and effective.
>
> To validate the effectiveness of our BOP in handling other types of 2D data, we conducted 'dimension reduction → reconstruction' experiments on the SpeechCommands dataset (composed of audio spectrograms) and the CWRU dataset (composed of spectrograms). Specifically, we first fed the data into BOP, PCA, or ICA to obtain low-dimensional features, and then trained an MLP to reconstruct signals from these low-dimensional features. As shown in the table below, BOP achieves substantially lower reconstruction error than PCA and ICA. These results demonstrate that our BOP possesses strong generalizability to various 2D data.
>
>
>
> | Algorithm  | Dataset        | Data Format        | MSE        | STD        | MAE        | STD        |
> | ---------- | -------------- | ------------------ | ---------- | ---------- | ---------- | ---------- |
> | PCA        | SpeechCommands | Audio Spectrograms | 0.0962     | 0.0008     | 0.2122     | 0.0016     |
> | ICA        | SpeechCommands | Audio Spectrograms | 0.1123     | 0.0014     | 0.2365     | 0.0031     |
> | BOP (Ours) | SpeechCommands | Audio Spectrograms | **0.0873** | **0.0007** | **0.2047** | **0.0015** |
> | PCA        | CWRU           | Spectrograms       | 0.1756     | 0.0058     | 0.3142     | 0.0077     |
> | ICA        | CWRU           | Spectrograms       | 0.1860     | 0.0071     | 0.3272     | 0.0087     |
> | BOP (Ours) | CWRU           | Spectrograms       | **0.1694** | **0.0039** | **0.3080** | **0.0042** |

---

> ### Author Response · Authors · 2025-11-19
> **Response to Reviewer UwUE, part 2**
>
> Q3: How significant is the computational overhead incurred by using both teacher and student networks, especially when compared to alternative approaches? Could the authors provide quantitative estimates or benchmarks to clarify this aspect?
>
> RQ3: Thank you for your question. We provide the **computational overhead incurred by both teacher and student networks** in the table below. We find that FACTS requires substantially fewer FLoating-point OPerations (FLOPs) and training time (in hours, H) compared with other multimodal methods, while achieving the best performance. In addition, we added the computational costs of each method and the corresponding analysis in  Appendix C.4 of the revised manuscript.
>
> | Model            | Dataset | FLOPs (Giga) | Training Time (H) |    MSE     |   STD    |    MAE     |   STD    |
> | :--------------- | ------- | :----------: | :---------------: | :--------: | :------: | :--------: | :------: |
> | AimTS            | Folsom  |     5.34     |       3.65        |   0.1366   |   1.26   |   0.1843   |   1.62   |
> | TimeVLM          | Folsom  |    31.03     |        9.7        |   0.1210   |   0.15   |   0.1599   |   0.24   |
> | FACTS (Proposed) | Folsom  |     0.93     |       1.54        | **0.0716** | **0.03** | **0.0968** | **0.05** |
> |                  |         |              |                   |            |          |            |          |
>
> Q4: The paper claims that random modality dropout enhances robustness when certain modalities are missing at inference time. Have the authors conducted experiments under more extreme or persistent missing-modality scenarios (e.g., complete loss of image data throughout inference)? Such evaluation would provide stronger evidence for the model’s robustness and practical deployability in real-world settings with unreliable sensors.
>
> RQ4: We appreciate this critical question regarding robustness. **We indeed conducted the challenging experiments you described, and the results are shown in Appendix C.3 and Table A-5.** We evaluated the extreme case where one or more auxiliary modalities are completely lost during inference. The results show that the model trained with Random Modality Dropout (RMD) experiences only a slight performance decrease (MSE increases from 0.0716 to 0.0937), while the model trained without RMD suffers a catastrophic performance collapse (MSE deteriorates from 0.0716 to 0.3553). These findings strongly confirm the robustness of our FACTS to modality missing.
>
> Q5: Is the ratio 𝜆 between the MSE loss and the CPCD loss a manually set hyperparameter or a learnable parameter? What is the value of this parameter, and how does varying 𝜆 affect the final model performance?
>
> RQ5: Thank you for the question. The hyperparameter $\lambda$ is manually set to 0.1 across all datasets.  As detailed in the sensitivity analysis in Section 4.4, the performance of FACTS remains stable across a wide range of $\lambda$ values (from 0.001 to 1.0), and the model performs well across the datasets covering distinct domains when $\lambda = 0.1$. This indicates that FACTS is robust to the variations of $\lambda$, which means that $\lambda$ is easy to tune.

---

> > ### Comment · Reviewer_UwUE · 2025-11-27
> >
> > Thank you for your reply, which has addressed my main concern. Overall, this is a basically qualified paper, and I will keep my init score.

---

> > > ### Author Response · Authors · 2025-11-27
> > >
> > > Dear Reviewer UwUE,
> > > Thank you for your feedback and for confirming that your concerns have been addressed. We appreciate your time and evaluation of our work.
> > > Best regards,
> > > The Authors

---

### Official Review · Reviewer_LJVM · 2025-10-31

**Soundness:** 3
**Presentation:** 2
**Contribution:** 2
**Rating:** 4
**Confidence:** 2

**Summary:**

The authors describe a method for inferring causal information related to time series by distilling a larger teacher model that does not require direct observations at test times. The main design choices of the architecture are to allow heterogeneous multimodal data to influence the time series prediction. The authors illustrate the performance of the method on a series of data sets and indicate that the proposed method outperforms the baselines.

**Strengths:**

The model empirically works very well and was tested on a variety of challenging datasets.

The authors consider a challenging problem and use a variety of techniques to propose a solution.

**Weaknesses:**

There does not seem to be a cohesive underlying reasoning for the improvement in performance. The paper introduces a number of known techniques in combination with their particular problem to indicate improvements in performance. However, it’s not a unifying theme underlying all of the changes the authors include to the model.

There do not seem to be any guarantees to ensure that the method is able to accurately disentangle the causal relationships, which is the main motivator of the paper.

**Questions:**

Did the authors consider any tests to see how well orthogonality is preserved during training?

Are there any specific guarantees one can make on the performance under perturbations such as spurious correlations? It would be interesting to see if the loss can be analyzed such that one can understand the conditions under which the model will be robust to such artifacts.

Under what conditions on the data can the model disentangle the underlying causal factors? Maybe analyzing this could be helpful to making stronger claims regarding the expected model performance.

---

> ### Author Response · Authors · 2025-11-19
> **Response to Reviewer LJVM, part 1**
>
> W1: There does not seem to be a cohesive underlying reasoning for the improvement in performance. The paper introduces a number of known techniques in combination with their particular problem to indicate improvements in performance. However, it’s not a unifying theme underlying all of the changes the authors include to the model.
>
> RW1: We appreciate the reviewer’s comment. **The unified theme of our FACTS is to leverage auxiliary modalities through a teacher network during training to uncover causal drivers, and then transfer this knowledge to a deployable student network that relies solely on historical data, thereby enhancing the student’s performance and reliability.** To achieve this, a Bilinear Orthogonal Projector (BOP) resolves structural and dimensional mismatches, enabling teacher and student networks to effectively and efficiently utilize auxiliary data using a unified backbone; Causal-Perturbation Contrastive Distillation (CPCD) enables student network **to separate the causal signals that are related to future time series** from perturbed pseudo signals; Lag-Aware Fusion and random modality dropout ensure that student network remains robust under real-world temporal misalignment and modality-missing. Ablation studies (see Table 2) show that replacing or removing these components significantly degrades the model performance, indicating that they are not simply combined but are complementary. Through these interlocking designs, FACTS consistently outperforms baseline methods on multiple real-world multimodal datasets, reducing MSE by 32.98% and MAE by 22.25% on average.
>
> W2: There do not seem to be any guarantees to ensure that the method is able to accurately disentangle the causal relationships, which is the main motivator of the paper.
>
> RW2: Thank you for raising the insightful question regarding causality. First, we would like to clarify that while strict theoretical causal disentanglement is an important research direction, our work **adopts a more practical objective**: **to separate 'faithful causal knowledge' from 'spurious statistical correlations' by leveraging known causal drivers (such as sky images).** We accomplish this through CPCD. During training, teacher network receives unperturbed inputs (historical data + true future auxiliary data) and perturbed inputs (historical data + incorrect future auxiliary data). Then, CPCD encourages the representations of student network to get close to the unperturbed features while moving away from the perturbed features.
>
> To intuitively validate the effectiveness of CPCD, we conducted a t-SNE visualization on the test set of Folsom dataset in Appendix C.6 of the revised manuscript. The teacher network’s unperturbed features (red) and perturbed features (green) correspond to two clearly separated trajectories, while the student network’s features (blue) largely overlap with the distribution of the unperturbed features (red). This demonstrates that CPCD successfully guides student network toward 'faithful causal knowledge' while moving it away from 'spurious statistical correlations', and thus achieving effective causal disentanglement. Moreover, the ablation results in Table 2 (row `CPCD') also support this claim, showing that replacing CPCD with other advanced distillation methods (e.g., CRD, TimeKD) substantially degrades performance.

---

> ### Author Response · Authors · 2025-11-19
> **Response to Reviewer LJVM, part 2**
>
> Q1: Did the authors consider any tests to see how well orthogonality is preserved during training?
>
> RQ1: Yes, **we would like to clarify that we already incorporated an orthogonality regularization** ($||U^{\top}U-I||_{F}^{2}+||V^{\top}V-I||_{F}^{2}$) to constrain the projection matrices in BOP to achieve orthogonality, as described in Section 3.1. In addition, we tracked the orthogonality loss throughout training. The loss values for the first 10 epochs are 1.342×10⁻³, 1.275×10⁻³, 1.165×10⁻³, 1.048×10⁻³, 9.43×10⁻⁴, 8.52×10⁻⁴, 7.75×10⁻⁴, 7.07×10⁻⁴, 6.48×10⁻⁴, and 5.98×10⁻⁴. These values exhibit a stable and monotonic decrease, indicating that the regularization effectively maintains the near-orthogonality of the projection matrices.
>
> Q2: Are there any specific guarantees one can make on the performance under perturbations such as spurious correlations? It would be interesting to see if the loss can be analyzed such that one can understand the conditions under which the model will be robust to such artifacts.
>
> RQ2: Thank you for the valuable comment. In our FACTS, **we achieve robustness to spurious correlations via minimizing $\mathcal{L}_{\text{CPCD}}$**, which is a contrastive loss. Mathematically, minimizing $\mathcal{L}_{\text{CPCD}}$ drives the student features $\mathbf{h}_S$ to align with the unperturbed teacher features $\mathbf{h}_T$ (representing true causal signals) while repelling them from the perturbed teacher features $\tilde{\mathbf{h}}_T$ (representing explicitly injected spurious correlations). **This 'align–repel' mechanism imposes a robustness constraint**, encouraging student network to learn a clear separation boundary in the feature space. **The effectiveness of this constraint hinges on the teacher network’s ability to distinguish between the 'true future' and the 'fabricated future'**, meaning that $\mathbf{h}_T$ and $\tilde{\mathbf{h}}_T$ must be separable in feature space. We empirically validate this separability through t-SNE visualizations (see Appendix C.6).
>
> Q3: Under what conditions on the data can the model disentangle the underlying causal factors? Maybe analyzing this could be helpful to making stronger claims regarding the expected model performance.
> RQ3: Thank you for raising this insightful question. The success of our model relies on two key data conditions:
>
> 1. Future auxiliary data (e.g., sky images) contains true causal drivers that reveal temporal dynamics. This condition is validated by the results in Table 1, where teacher network (with access to future auxiliary data) consistently outperforms student network (which relies solely on historical data). This demonstrates that the future auxiliary data indeed carries essential causal information.
> 2. The perturbed examples (with random future data) can be clearly distinguished from the unperturbed examples (with true future data). This condition is supported by the visualization results in Appendix C.6 of the revised manuscript, which show a clear separation between the unperturbed (red) and perturbed (green) features.
>
> We have elaborated on these 'data conditions' in Appendix C.5 of the revised manuscript, as they are essential for understanding the applicability and effectiveness of our FACTS framework.

---

> ### Author Response · Authors · 2025-11-28
> **Gentle Reminder regarding our Rebuttal (Submission22522)**
>
> Dear Reviewer LJVM,
>
> We sincerely appreciate your time and the constructive feedback provided. We have made every effort to address the concerns in reviews, particularly regarding the model's underlying reasoning (W1) and the causal disentanglement (W2/Q2). We hope that our response, along with the new visualization results and clarifications, has adequately answered your questions.
>
> Given that the author-reviewer discussion window is closing soon, could you please let us know if you have any further feedback? We are standing by to address any issues immediately.
>
> Thank you for your time and consideration.
>
> Best regards,
>
> The Authors

---

### Official Review · Reviewer_o6rZ · 2025-11-01

**Soundness:** 3
**Presentation:** 3
**Contribution:** 3
**Rating:** 6
**Confidence:** 3

**Summary:**

The paper proposes FACTS, a teacher-student framework designed to address multimodal time series forecasting by leveraging auxiliary modalities to capture causal drivers of temporal dynamics. The core innovation lies in using a teacher network with access to future auxiliary data during training to distill causal knowledge into a deployable student network that uses only historical data. The authors test FACTS on four datasets and report performance improvements over baselines.

**Strengths:**

1. Articulates why unimodal forecasting fails and why auxiliary modalities capture critical causal drivers.
2. Using perturbed future auxiliary data as negative samples in contrastive distillation is intuitive and aligns with causal learning principles.
3. Compares against methods spanning unimodal, LLM-based, and multimodal approaches with systematic ablations.
4. Outperforms baselines on various datasets with reduced standard deviations, suggesting robustness.

**Weaknesses:**

1. Teacher-student distillation, contrastive learning, and bilinear factorization are well-established; the contribution is primarily architectural integration rather than methodological innovation.
2. The author emphasized the multimodal, but only compared three multimodal datasets.
3. No justification for why δ_max = {0,1,...,5} works across datasets.
4. Many hyperparameters are introduced; how to keep the fairness of these hyperparameters across different datasets?

**Questions:**

See above

**Details Of Ethics Concerns:**

The author did not use an anonymous GitHub repository, which may potentially expose identifying information. This is not a major issue, but it should have been mentioned.

---

> ### Author Response · Authors · 2025-11-19
> **Response to Reviewer o6rZ, part 1**
>
> W1: Teacher-student distillation, contrastive learning, and bilinear factorization are well-established; the contribution is primarily architectural integration rather than methodological innovation.
>
> RW1: Thanks for the valuable comment. We agree that the teacher–student distillation, contrastive learning, and bilinear factorization are well-established. However, rather than simply integrating them, we develop specialized algorithms with methodological innovations to address the key challenges under the novel setting of multimodal Time Series Forecasting (TSF). Specifically, to address data heterogeneity and mismatch, we design the Bilinear Orthogonal Projector (BOP) to transform high-dimensional auxiliary data into low-dimensional temporal representations **in an end-to-end manner**. To learn useful cross-modal causal information, we build a multimodal teacher–student framework with Causal Perturbation Contrastive Distillation (CPCD), **which explores the causal relations from future auxiliary data** during training and transfers them to a student network that relies only on historical data. In summary, these innovations are closely coupled and work together to learn a precise and reliable student network for multimodal TSF.
>
> W2: The author emphasized the multimodal, but only compared three multimodal datasets.
>
> RW2: Thanks for the reviewer’s comment. First, we would like to clarify that our FACTS is actually evaluated on **four** real-world multimodal datasets (Folsom, SKIPP’D, CCG, and CRNN), which include **two distinct application domains** (solar power generation and water-level monitoring).
>
> Importantly, our work focuses on a challenging and less-explored multimodal TSF setting, where **each timestamp is paired with its own auxiliary data**. This is fundamentally different from previous multimodal TSF approaches [1, 2], which **associate an entire time series with a single static image or text description**. Admittedly, obtaining such time-varying auxiliary data requires dedicated sensing infrastructure (e.g., high-frequency sky cameras) and long-term continuous monitoring. Therefore, there are very few publicly available datasets meeting this requirement.
>
> Finally, our FACTS consistently performs well across four multimodal datasets, which indicates that FACTS is not tailored to a particular dataset but generalizes across diverse domains and modality configurations.
>
> W3: No justification for why δ_max = {0,1,...,5} works across datasets.
>
> RW3: Thank you for the valuable comment. **$\delta_{max}=5$ is a scale-invariant discrete search window**, and its corresponding absolute time span depends on data sampling frequency (e.g., 5 minutes, 15 minutes, or 1 hour). The sampling frequencies on different datasets are **meaningfully chosen** to capture the key dynamics of each specific physical system [3, 4]. In the systems sampled at **these frequencies**, the informative initial causal lags usually emerge within only a few discrete time steps. Therefore, a unified and suitable $\delta_{max}$ configuration can work across different datasets.
>
> In our FACTS framework, **we set $\delta_{max}=5$ to balance lag-capturing capability and computational efficiency,** as discussed in Section 4.4. Specifically, a small window cannot capture essential temporal misalignments between modalities, while a large window would introduce unnecessary computational overhead. To further validate this setting, we conduct a sensitivity analysis of $\delta_{max}$ across datasets, as shown in the table below. The results show that performance improves as $\delta_{max}$ increases but stabilizes once it reaches 5, which indicates that $\delta_{max}=5$ provides an effective and efficient lag window.
>
> | Dataset | $\delta_{max}=1$ |        | $\delta_{max}=3$ |        | $\delta_{max}=5$ |        | $\delta_{max}=7$ |        |
> | ------- | ---------------- | ------ | ---------------- | ------ | ---------------- | ------ | ---------------- | ------ |
> |         | MSE              | MAE    | MSE              | MAE    | MSE              | MAE    | MSE              | MAE    |
> | Folsom  | 0.0811           | 0.1199 | 0.0775           | 0.1089 | 0.0716           | 0.0968 | 0.0713           | 0.0975 |
> | SKIPP'D | 0.3582           | 0.4617 | 0.3275           | 0.4342 | 0.2876           | 0.3843 | 0.2872           | 0.3836 |
> | CCG     | 0.0036           | 0.0384 | 0.0031           | 0.0338 | 0.0028           | 0.0315 | 0.0029           | 0.0327 |
> | CRNN    | 0.1173           | 0.2771 | 0.1148           | 0.2615 | 0.1121           | 0.2497 | 0.1121           | 0.2499 |

---

> ### Author Response · Authors · 2025-11-19
> **Response to Reviewer o6rZ, part 2**
>
> W4: Many hyperparameters are introduced; how to keep the fairness of these hyperparameters across different datasets?
>
> RW4: We thank the reviewer for the valuable comment regarding fairness. Although our FACTS introduces several key hyperparameters, **we intentionally avoid dataset-specific fine-tuning and employ a unified set of hyperparameters across all datasets to ensure a fair comparison**.
>
> Specifically, in our FACTS, $\lambda$ balances the contributions of MSE loss and CPCD loss, $r_{h}$ and $r_{w}$ in BOP determine the degree of data compression, and $\delta_{\max}$ controls the lag window size. As analyzed in Section 4.4, FACTS is robust to the variations of $\lambda$ and consistently yields competitive performance across datasets when $\lambda=0.1$, so we adopt it as the default setting. Similar to the configuration of $\delta_{max}=5$ detailed in RW3, we adopt $r_{h}=r_{w}=8$ to balance information preservation and computational efficiency, where a small $r_{h}$ (or $r_{w}$) results in large information loss and a large value would incur substantially higher computational cost.
>
> As shown in Figure 3, our FACTS with the above hyperparameter settings achieves satisfactory performance on the datasets across distinct domains. This demonstrates that our model’s superior **performance arises from its architecture and learning mechanisms rather than dataset-specific parameter tuning**. For baseline methods, we follow the settings provided in their official implementations to ensure a fair comparison.
>
> W5: The author did not use an anonymous GitHub repository, which may potentially expose identifying information. This is not a major issue, but it should have been mentioned.
>
> RW5: We thank the reviewer for raising the concern regarding code anonymity. In our initial submission, we provided the repository using an anonymous GitHub account. To further enhance the anonymization and address the reviewer's concern, we have now migrated the repository to anonymous.4open.science and have updated the link (https://anonymous.4open.science/r/FACTS-7F94) in the revised manuscript.
>
> ###### References
>
> [1] Zhong et al., Time-VLM: Exploring Multimodal Vision-Language Models for Augmented Time Series Forecasting, ICML, 2025.
>
> [2] Chen et al., AimTS: Augmented Series and Image Contrastive Learning for Time Series Classification, ICDE, 2025.
>
> [3] Mikofski et al., Effects of Solar Resource Sampling Rate and Averaging Interval on Hourly Irradiance Data Quality, NREL Technical Report, 2023.
>
> [4] Bruel et al., Monitoring Aquatic Ecosystems: The Role of Sampling Frequency in Capturing Environmental Dynamics, Ecological Indicators, 2019.

---

> > ### Comment · Reviewer_o6rZ · 2025-11-28
> >
> > Thanks for the authors' responses, and I will keep my positive score.

---

> > > ### Author Response · Authors · 2025-11-28
> > >
> > > Dear Reviewer o6rZ,
> > >
> > > Thank you for reviewing our response and for maintaining the positive score.
> > >
> > > Best regards,
> > >
> > > The Authors

---

### Author Response · Authors · 2025-11-25
**Follow-up on our Rebuttal and Revisions**

**Dear AC and Reviewers,**

We hope this message finds you well.

We are writing to respectfully follow up on our rebuttal posted on **November 20th**. As the discussion period progresses, we want to ensure that our detailed responses have effectively addressed your concerns.

- In our revision, we have focused on the following key updates:
  - **Reviewer LJVM:** Validated **Causal Disentanglement** via t-SNE visualizations and clarified the unified architectural logic.
  - **Reviewer UwUE:** Verified **BOP Generality** and confirmed **Computational Efficiency** of our method.
  - **Reviewer o6rZ:** Clarified **Methodological Innovations** and justified **Hyperparameter Settings**.

We would greatly appreciate your feedback on these updates. We remain fully available to conduct further clarifications before the discussion phase concludes.

Thank you again for your time and dedication to improving our paper.

Sincerely,

The Authors

---

### Author Response · Authors · 2025-12-02
**Rebuttal Summary for Submission #22522**

**Dear Area Chair**,

To assist with your review process, we provide a concise summary of the current status and key updates made during the discussion phase.

**1. Reviewer Consensus & Status**

**Reviewers UwUE (Scores: 6, Positive) & o6rZ (Scores: 6, Positive):** Both reviewers have explicitly confirmed that our responses addressed their concerns (regarding generalizability, efficiency, and hyperparameters), and they have decided to maintain their positive scores.

**Reviewer LJVM (Score: 4, Borderline):** The reviewer raised concerns about causal disentanglement. In our revision, we provided detailed methodological clarifications alongside t-SNE visualizations to demonstrate the separation of true causal drivers from spurious correlations. We are standing by for any further feedback.


**2. Key Rebuttal Actions**

**Proven Generality (Response to UwUE):** We extended our proposed Bilinear Orthogonal Projector (BOP) to non-image domains. New experiments on audio spectrograms (SpeechCommands) and spectrograms (CWRU) show BOP significantly outperforms PCA and ICA.

**Causal Disentanglement Verification (Response to LJVM):** We clarified the "align-repel" mechanism of our method and added t-SNE visualizations (Appendix C.6). These show a clear boundary between "true causal features" and "spurious features," confirming the student network successfully captures the former.

**Efficiency & Robustness (Response to UwUE & o6rZ):** We provided a FLOPs table (0.93G vs. TimeVLM's 31.03G) to demonstrate efficiency, and justified the hyperparameter settings (e.g., lag window $\delta_{max}$) based on physical system dynamics and sensitivity analysis.

Thank you for your time and effort.

Best regards,

The Authors

---

### Meta-Review · Area_Chair_sGa3 · 2026-01-06

**Summary:**

This paper proposes FACTS, a Future-Aided Causal Teacher-Student framework for multimodal time series forecasting (mainly time series + image), , where a teacher leverages future auxiliary modalities during training and distills knowledge into a deployable student that uses only historical inputs at test time.

The authors have made tremendous efforts for the rebuttal. I have carefully read the paper, the reviews, and the rebuttals. I agree with Reviewer LJVM that “The paper introduces a number of known techniques incombination with their particular problem to indicate improvements in performance.” It is really difficult to tell what is the real technical contribution or what technique is really working, among the proposed methods, Causal-Perturbation Contrastive Distillation (CPCD), Multimodal Data Fusion, Backbone with Bidirectional Attention, and Bilinear Orthogonal Projector (BOP). While the authors include ablations (Table 2), these experiments do not sufficiently disentangle the effects of these ingredients, nor do they convincingly address the key concern about the method’s actual novelty and necessity of each module. More comprehensive interpretability studies would be necessary to clarify the individual contributions and necessity of each component.

Additionally, I find that the current title does not clearly or accurately summarize the main technical ideas of the work, which further contributes to the ambiguity around the paper’s central contribution.

Therefore, I think this paper is less than a rigorously justified ICLR-level contribution, thus I recommend rejection.

**Reviewer Concerns:**

Some presentation issues and experimental details were clarified during the rebuttal. However, the core technical contribution remains insufficiently clear, and it is still difficult to pinpoint which specific technique is responsible for the reported performance gains.

**Reviewer Scores:**

All three reviewers decided to maintan their initial scores (6,6,4) during the rebuttal. I also think they will maintain their scores.

---

### Decision · Program_Chairs · 2026-01-26

Reject